# Vesicle and reaction-diffusion hybrid modeling with STEPS
Iain Hepburn ⓘ , Jules Lallouette ⓘ , Weiliang Chen ⓘ , Andrew R. Gallimore, Sarah Y. Nagasawa-Soeda &
Erik De Schutter ⓘ ✉

Vesicles carry out many essential functions within cells through the processes of endocytosis, exocytosis, and passive and active transport. This includes transporting and delivering molecules between different parts of the cell, and storing and releasing neurotransmitters in neurons. To date, computational simulation of these key biological players has been rather limited and has not advanced at the same pace as other aspects of cell modeling, restricting the realism of computational models. We describe a general vesicle modeling tool that has been designed for wide application to a variety of cell models, implemented within our software STochastic Engine for Pathway Simulation (STEPS), a stochastic reaction-diffusion simulator that supports realistic reconstructions of cell tissue in tetrahedral meshes. The implementation is validated in an extensive test suite, parallel performance is demonstrated in a realistic synaptic bouton model, and example models are visualized in a Blender extension module.

Vesicles play many diverse and essential roles in cell biology. For example in neurons, synaptic vesicles uptake and store neurotransmitter, releasing the contents into extracellular space upon a presynaptic signal[1], and post-synaptic endocytosis and exocytosis of vesicles containing AMPA-type glutamate receptors (AMPARs) determine expression of AMPARs in the cell surface[2,3] via the vesicular endosomal pathway[4]. Therefore vesicles play central roles both in chemical communication between neurons and in determining the strength of synaptic contacts, two vital processes of brain function. Vesicles diverse in structure, molecular composition (even within functionally similar systems across anatomies, such as in the case of synaptic endocytosis[5]), and function play many more important roles throughout cell biology and are essential to life processes.

The field of computational biology has turned its attention to modeling these key cell biology players in recent years to varying degrees of biological detail. Many of these early applications are model-specific with a focus on one particular aspect of vesicle function that is being investigated in the study often alongside an experimental observation. Examples of this include an investigation of autophagy[6], vesicle mobility in the presynaptic space[7,8], calcium-mediated release probability on docked synaptic vesicles[9], and the influence of cytoskeletal density on vesicle diffusivity[10]. A stochastic model of post-synaptic AMPAR trafficking by Tanaka and colleagues[11] is one of the most detailed vesicle models to date and includes many of the key features of the system such as endocytosis and AMPAR trafficking, but does have some simplifications and limitations such as drawing docking and exocytosis from random distributions instead of modeling full diffusion and with limited molecular complexity.

Whilst these early studies have paved a path to computational vesicle modeling, there is a need for a general tool or tools that have the flexibility to enable application to a variety of modeling systems. In addition, such a tool should also allow modeling of other features of the biological environment such as molecular reaction-diffusion processes and voltage-gated channels alongside the vesicle dynamics, in order to enable full modeling of the rich chemical processes behind important vesicle functions such as docking, priming and fusion[12].

In terms of designing a tool for such computational vesicle modeling, one possible approach is to model the forces that vesicles experience in the cytosolic environment explicitly by their interactions with proteins and other molecules in their environment (including water) to atomic detail, thus extending the field of molecular dynamics[13]. This approach has the benefit of high morphological and physical realism, however the drawback is that the biological time that can be achieved in simulations is severely limited on current hardware due to the high intensity of computation, and often the experimental data of such atomic detail is absent[14]. Although coarse-grained approaches can increase the achievable biological time somewhat to the order of nanoseconds or microseconds[15], this still puts out of reach many models of interest in vesicle systems that may operate on a scale orders of magnitude higher such as synaptic vesicle endocytosis that operates on a timescale of seconds[16] within a synaptic vesicle cycle operating on a timescale of minutes[17].

In order to achieve these biological times, a good approach is to extend one of the current tools for subcellular modeling, bringing the many benefits of these established tools but with a vesicle-specific extension. STochastic

Computational Neuroscience Unit, Okinawa Institute of Science and Technology, 1919-1 Tancha, Onna-son, Okinawa, Japan. ✉e-mail: erik@oist.jp

Engine for Pathway Simulation (STEPS) is a voxel-based simulator that extends Gillespie's Stochastic Simulation Algorithm (SSA)[18] to simulate diffusion within tetrahedral meshes that are able to capture complex biological boundaries accurately[19]. STEPS has been extended since its initial release to support voltage calculations on the tetrahedral mesh[20] and parallel computation based on the Message Passing Interface (MPI) protocol[21] via an operator-splitting framework[22]. The main benefit of SSA approaches is that they are fast and scalable[21], and capture reaction-diffusion kinetics accurately as long as certain geometrical criteria are met[19,23]. Although STEPS was originally designed with a focus on neural systems such as simulating molecular models of synaptic plasticity[24–26] or other dendritic spine molecular models[27], along with larger-scale dendritic systems[28–30], it has also been applied to other areas of biology such as to model viral RNA degradation and diffusion[31], astrocytic calcium signaling[32] and nanoparticle penetration in cancer research drug discovery[33]. The power of the parallel solver within STEPS was recently demonstrated by simulation of an entire Purkinje cell at molecular detail for seconds of biological time on a supercomputer[34].

In this study we describe how we take the many benefits of this parallel SSA-based simulator and extend it for modeling vesicles, which cannot realistically be represented by point-molecules within the regular SSA. Our aim is to provide a flexible modeling tool that captures many important features of vesicles and their processes so that the tool can be applied to many different model systems. Vesicles are represented with enough realism to extract their key features such as mobility within cells and near membranes, their molecular interactions, and important features such as endocytosis and exocytosis, but by an approach that is capable of achieving biological times that are out of reach for molecular dynamics approaches. We describe our unique hybrid SSA-vesicle approach where vesicles and vesicle-related phenomena are given special treatment but with a firm focus on computational performance, and the benefits of the SSA are maintained for other chemical species and reaction-diffusion processes. We describe in some detail the implementation of our methods, and demonstrate biological plausibility by an extensive validation suite of all the new modeling components introduced in this paper. In addition, we discuss potential sources of error in models and how these can be avoided or minimized. We introduce the parallel MPI-based implementation, demonstrate performance in a realistic synaptic bouton model, and visualize in a Blender extension module.

## Results

In this section we describe sequentially the features of vesicles that are modeled, and with results from test models by means of validating the method. We begin by qualitatively exploring vesicle mobility and crowding effects in the STEPS implementation. Later validations, however, are then generally run under well-mixed conditions (where crowding effects are negligible) in order to match to analytical solutions, although we should note in more complex models the reaction rates can be under-sampled or over-sampled due to effects of crowding.

Figure 1 shows schematically the biological phenomena that STEPS simulates in the vesicle extension. Each of these features are described within the following sections.

### Vesicle overlap with tetrahedral elements in a mesh

Vesicles exist within the tetrahedral mesh environment that represents cellular volumes such as the cytosol, and vesicles may not exist outside of the mesh boundaries. Based on the position and size of the vesicles, the overlap between vesicles and the tetrahedrons is found (see Sphere-Tetrahedron overlap) and each individual vesicle as a computational unit stores information about this overlap. Each vesicle knows which tetrahedrons it overlaps, and for each tetrahedron, the volume of the tetrahedron that it overlaps, which allows all the features presented in this study to be supported such as interaction with the SSA, passive transport, and so on. As we aim to provide a general modeling tool to support a wide range of modeling scenarios there is no restriction on the size of this overlap, however Fig. 2 gives information about the range of overlap from the meshes used in the study, covering a wide range of modeling scenarios. From the largest tetrahedrons that average over 100 nm in size

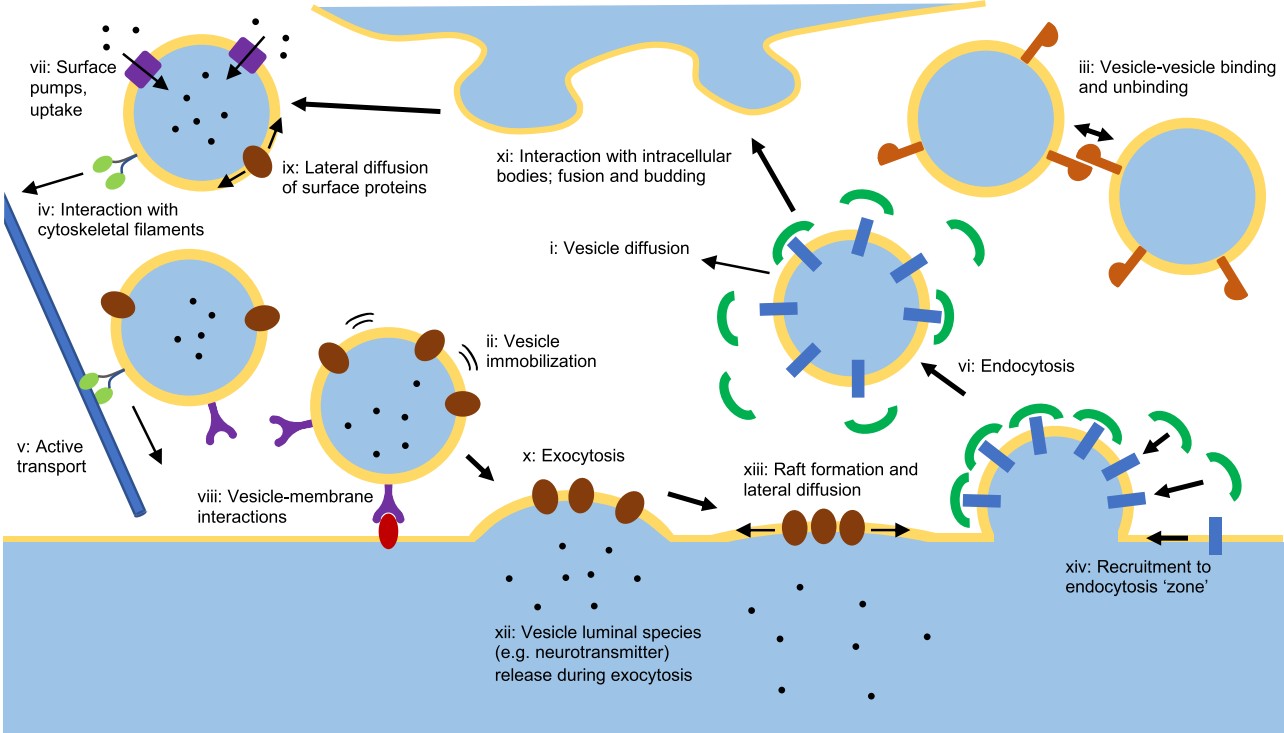

**Fig. 1 | The full set of behaviors that STEPS implements for Vesicle and Raft modeling, described further in text.** As such, models like the synaptic vesicle cycle including trafficking, docking, exocytosis and endocytosis are supported, as well as variations on those models such as kiss-and-run fusion and recycling. These vesicle models can exist alongside and interact with voxel-based reaction-diffusion and bioelectrical cellular models.

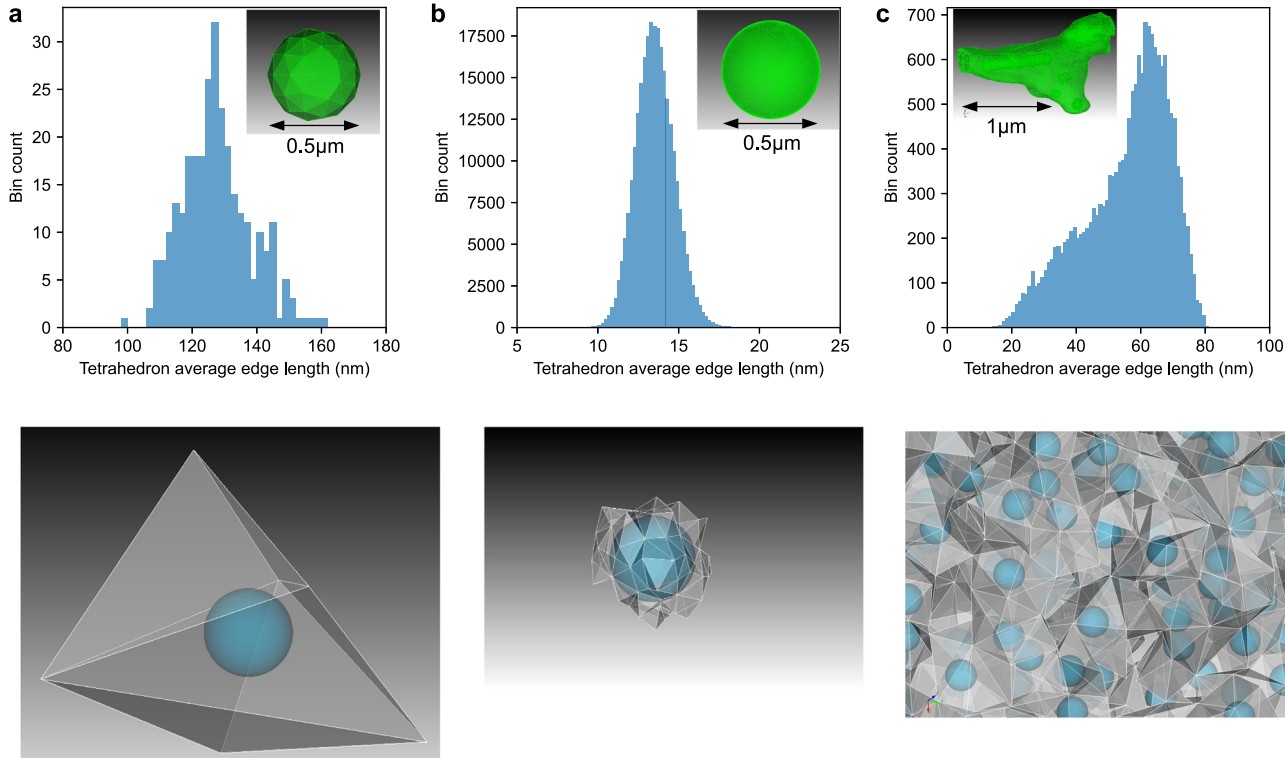

**Fig. 2 | Size comparison between vesicles and the tetrahedral meshes used in this study.** Each separate panel shows the binned average edge length of all tetra-hedrons for a mesh, along with, below, an example size comparison between a vesicle or vesicles of diameter 40 nm and the tetrahedrons that they overlap in the mesh. **a** A mesh of a spherical volume of 0.5 μm diameter comprised of just 291 tetrahedrons, which were the largest tetrahedrons that it was possible to generate for this mesh shape. **b** A mesh of the same spherical volume of 0.5 μm diameter comprised of 265,307 tetrahedrons, which are the smallest tetrahedrons used in this study. **c** The mesh from the realistic synaptic bouton model of Gallimore et al.[35], also used in this study.

(Fig. 2a), synaptic vesicles of 40 nm diameter typically overlap just one or a small number of tetrahedrons, whereas Fig. 2b shows a scenario where tetrahedron size is smaller than vesicle diameter and so vesicles overlap many tetrahedrons at any given time. For the most realistic modeling scenario presented in this study (based on a realistic synaptic bouton model[35]), tetrahedrons are of comparable size to synaptic vesicles (Fig. 2c). The modeling features are designed to work under any of these scenarios, though they can present different sources of errors, as is dis-cussed later in Vesicle surface protein and vesicle internal molecule transport, and interaction with environment.

### Vesicle mobility: passive and active transport in the crowded cell environment, partial mobility in clusters, and tethering to the cell membrane

One of the most important features of vesicles is that they exhibit various forms of mobility, dependent on many factors such as location, function and cell activity[36]. They can, for example, be freely-diffusing within the crowded environment of the cytosol or other cell compartments (Fig. 1(i)), effectively immobile when docked to tethering molecules in the cell membrane (Fig. 1(ii)), or partially mobile when clustered to other vesicles[37] in functional pools[38] (Fig. 1(iii)). STEPS supports all of these situations. By default, vesicles freely diffuse within their cellular boundaries by Brownian motion diffusion (see Vesicle definition and diffusion). In STEPS, this is modeled as changes in position on fixed, user-defined time steps, $\Delta t$, and so the size of these diffusion steps depends on diffusion coefficient and the size of $\Delta t$. Figure 3a shows example diffusion steps for a diffusion coefficient of 0.06 μm²s⁻¹ (which is the free cytoplasmic diffusion of synaptic vesicles after hydro-dynamic effects found in ref. 8 and also used in Fig. 3c) for typical $\Delta t$ choices of 0.1 ms and 1 ms. Comparison between Fig. 3a and Fig. 2c show that vesicles usually diffuse less than the size of one tetrahedron per diffusion step

under realistic modeling scenarios, although this is not a restriction on a STEPS model.

Because vesicles cannot overlap other vesicles, mesh boundaries, or other boundaries that might represent, for example, internal cellular structures, the mobility of vesicles is reduced in the presence of such structures[8] thus capturing the important known features of cellular crowding[39]. To demonstrate this, first we investigated vesicle diffusion in the case where 0%, 20%, 40% and 60% of the cell volume is occupied, showing how the software captures the expected diffusion distance in the 0% case, and mobility reduces with increasing volume occupancy (Fig. 3b), as expected. Supplementary Note 1 gives further information about this model.

In addition, we tested a mitochondria-based model introduced by Rothman et al.[8], as detailed in Supplementary Note 2. In this model 28% of the volume is occupied by immobile mitochondria in the tetrahedral mesh, a further 25% by immobile vesicles, and 17% by mobile vesicles (the model is shown in the inset of Fig. 3c). We demonstrate that STEPS captures the expected features of this model in that the effective diffusion rate of the mobile vesicles is not constant in time; it approximates the Brownian-motion rate (0.06 μm²s⁻¹ in this model) only on short time-scales and is reduced by steric interactions on longer timescales (Fig. 3c) approaching physiological values[8]. STEPS does not model the very fast timescale effects of hydrodynamics[8].

**Vesicle active transport.** Active transport on structures such as microtubules and actin filaments (Fig. 1(iv,v)) is an essential feature of vesicle mobility, especially in neurons[40]. For example, synaptic vesicle trafficking along actin filaments delivers vesicles to the active zone[41], and post-synaptically recycling vesicles are actively transported to the spine surface to deliver AMPARs during long-term potentiation[42]. Vesicles are transported by motor proteins myosin on actin, and kinesin and dynein

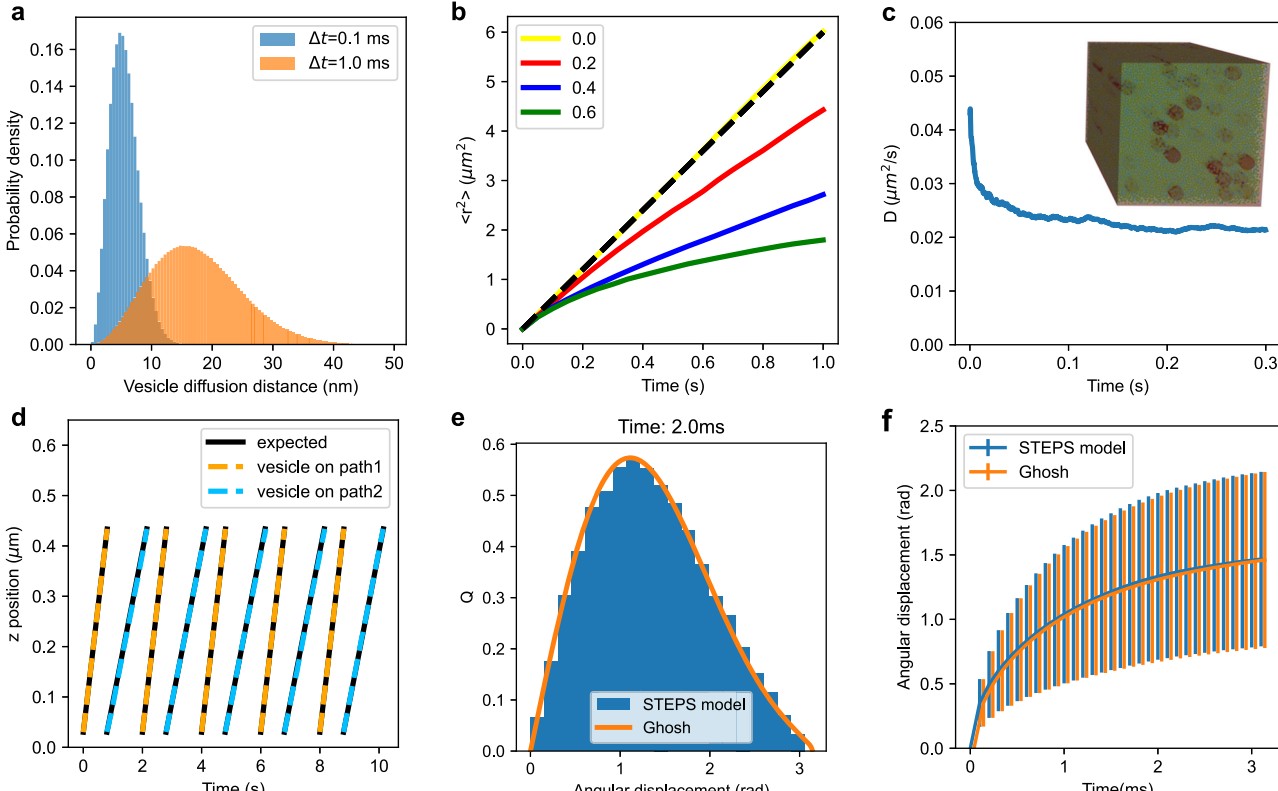

**Fig. 3 | Validation of vesicle mobility, and mobility of molecules on vesicle surface. a** Vesicle diffusive steps experienced by a vesicle diffusing at 0.06 μm²s⁻¹ for two different choices of $\Delta t$ of 0.1 ms (blue) and 1 ms (orange). **b** Vesicle diffusion in a cytosolic volume with varying fractions of the volume occupied. Free diffusion (yellow line) is compared to expected 6Dt solution, and increasing cytosolic occupancy of 20%, 40% and 60% show increasingly reduced mobility, as expected. **c** The apparent diffusion coefficient of vesicles vs time in the STEPS implementation of the MFT model of Rotham et al.[8] showing reduced apparent diffusion rate with time due to steric interactions with mitochondria and other vesicles, agreeing with the results

on microtubules[43]. A feature of the motor-protein walk is static periods punctuated by sharp steps of regular size to the next position along the structure[44–46]. The dwelltimes of which are near single-exponential or double-exponential[45,47,48], indicating one or sometimes two rate-limiting steps.

STEPS models active transport by a system of branched `Paths` representing cytoskeletal filaments that vesicles may interact with if they cross them in their environment. Vesicles travel along the path from the beginning point to the end point by a user-defined speed and stepsize (for example the stepsize of myosin V is 36 nm, consisting of a 25 nm working stroke and an 11 nm diffusive movement[49]). The default behavior is to sample dwelltimes from a single-exponential distribution, although a double-exponential may optionally be used. The total distance traveled divided by the sum of the dwelltimes then equals the given speed, with noise arising from the distribution of dwelltimes (instead of a single value). Supplementary Note 3 and Supplementary Fig. 1 demonstrate the implementation of single-exponential and double-exponential dwelltimes in STEPS and Supplementary Video 1 shows vesicles undergoing active transport on a path in STEPS in a simple model.

As well as active transport, paths may be useful in other modeling contexts such as vesicle binding to actin cytoskeleton in cluster formation[50] and active zone tethering[51].

We validate the implementation of paths in a model measuring vesicle displacement on path vs expected displacement, as demonstrated in Fig. 3d. The model vesicles undergo several periods of active transport along paths at the expected speeds (in this example with small stepsize to counteract

from that study. **d** Validation of active transport, with vesicles maintaining the correct speeds upon interactions with paths at 0.5 μm s⁻¹ (orange) and 0.3 μm s⁻¹ (blue). **e** The STEPS implementation (blue) of surface species angular displacement by the Ghosh algorithm[52] (orange) for an example diffusion step of 2 ms for a surface species with diffusion rate on the vesicle surface of 1 μm²s⁻¹. **f** Mean and standard deviation of spherical surface diffusion over a range of diffusion timesteps ($n$ = 100,000 independent simulations). The Ghosh algorithm (orange) is compared to a surface diffusion model on a spherical mesh surface in STEPS (blue).

stochastic effects for the purpose of validation). The implementation of this model is described in further detail in Supplementary Note 4.

**Vesicle surface molecule diffusion.** Vesicles contain proteins in their surface which may have been present at the site of endocytosis (Fig. 1(vi)) and which can undergo various interactions such as phosphorylation upon interaction with the cytosolic environment. These vesicle surface molecules may play important roles such as neurotransmitter uptake (Fig. 1(vii)), docking to cell membrane (Fig. 1(viii)) or binding to other vesicles (Fig. 1(iii)). These molecules may be mobile in the vesicle surface, undergoing lateral diffusion (Fig. 1(ix)). This amounts to modeling diffusion on a spherical surface, by which STEPS applies the efficient approximate propagator algorithm of Ghosh et al.[52]. As described further in Supplementary Note 5 and in Supplementary Fig. 2, to test the STEPS implementation of the Ghosh algorithm we tested a regular diffusion model on a spherical surface mesh in STEPS and compared to the Ghosh algorithm (Fig. 3e shows one example of angular displacement after time 2 ms of a surface molecule diffusing at 1 μm²s⁻¹). The Ghosh algorithm compares accurately to the measured angular displacement distribution from our surface diffusion model (Fig. 3f) and is a reliable, fast algorithm for implementing diffusion of surface molecules on vesicles in STEPS.

**Exocytosis, endocytosis, fusion and budding**

Exocytosis (Fig. 1(x)) is an essential cell process by which luminal species inside vesicles can be secreted into the extracellular space (for example neurotransmitter release at chemical synapses) and membrane proteins

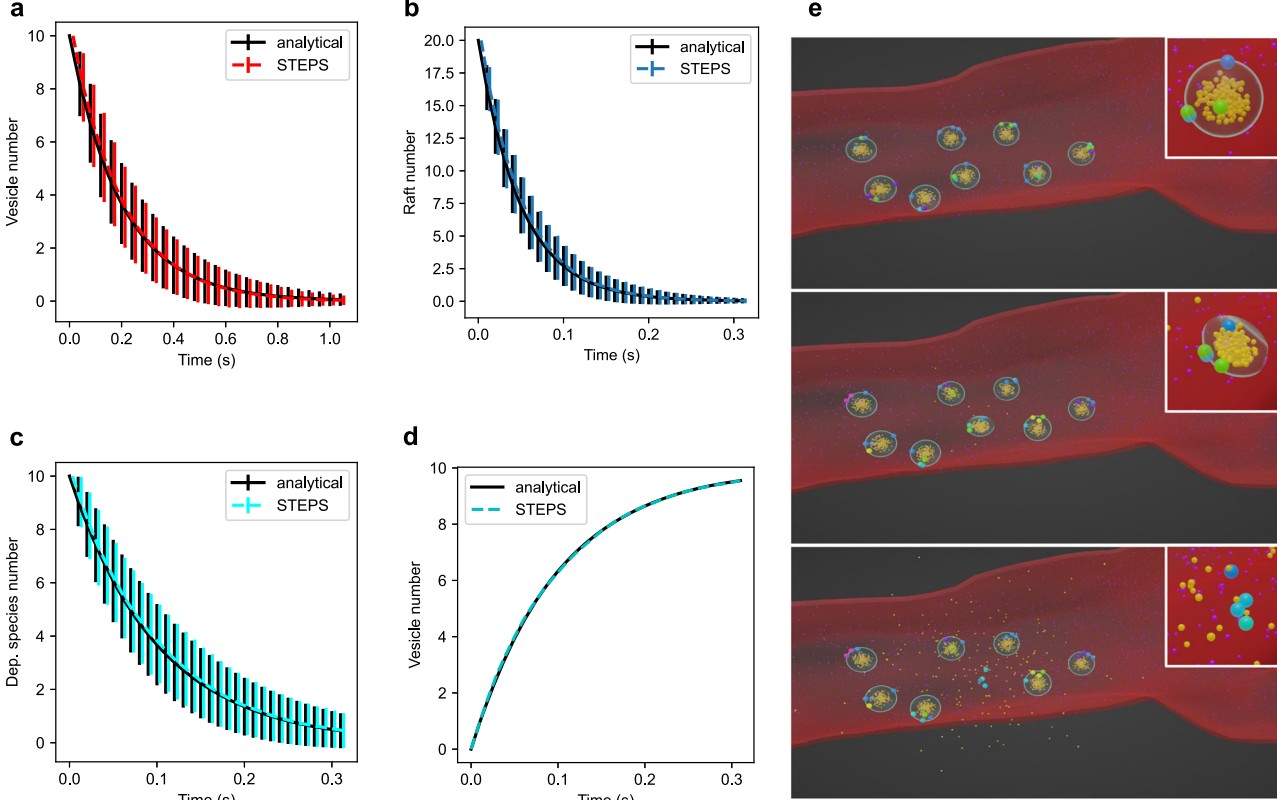

**Fig. 4 | Validation of Endocytosis and Exocytosis in STEPS by basic test models where analytical comparisons can be made. a** 10 vesicles undergo exocytosis at a rate of 5 s$^{-1}$ in a model where diffusion is not rate-limiting. Vesicle number decays exponentially, as expected. Mean and standard deviation of $n = 1000$ independent simulations with different random number sequences are shown to validate correct implementation within the SSA. **b** Raft endocytosis is similarly validated as an exponential decay of 20 Rafts in the model at a rate of 20 s$^{-1}$ (mean and std of $n = 1000$ independent simulations). Endocytosis at rate 10 s$^{-1}$ is validated as exponential decay of a dependent species on membrane where 1 molecule is internalized per endocytosis event (**c**, mean and std of $n = 1000$ independent simulations) and the

increase in vesicle number due to endocytosis (**d**). **e** An exocytosis event visualized in Blender. Top panel: vesicles are docked in the active zone by SNARE complexes and calcium is released in the cytosol (purple). Middle panel: The vesicle in the center starts to undergo exocytosis. Bottom panel: Exocytosis is complete, neurotransmitter (dark yellow) is released into the extracellular space and the SNARE complex remains in the membrane for dismantling. The different colors of the SNARE complexes indicate different stages of priming, with the fully primed state on which exocytosis depends shown in light blue. The inset in each panel shows a close-up (from a different angle) of the vesicle undergoing exocytosis.

may be inserted into the cell membrane. Both processes are vital to cell function. Vesicles are also able to fuse to internal membranes such as endosomes (Fig. 1(xi)), and both this and exocytosis are modeled in STEPS by the same computational framework within the SSA. STEPS is able to model the complex chemical processes leading up to and resulting in exocytosis, such as those involved in docking, priming and fusion (Fig. 1(viii))[53].

For synaptic vesicles it has been proposed that kiss-and-run fusion[54], whereby a vesicle does not completely collapse upon neurotransmitter release but remains intact and is immediately endocytosed and recycled, is utilized in neurons due to the clear advantage of speed of recycling. Full-collapse fusion and kiss-and-run are proposed to coexist in many systems, although the prevalence and significance of kiss-and-run fusion is somewhat debated (for reviews see refs. 55,56). In addition, another mode of release known as open and closed fusion has been described and, like kiss-and-run, does not result in full collapse of the vesicle but is distinct from kiss-and-run for example by amount of vesicle content released and is proposed as the main mode of exocytosis[57]. Partial release of luminal content during open and closed fusion has been directly measured, for example Li et al.[58] measure 64% of total catecholamine content release during stimulated exocytosis in PC12 cells. Since a modeler may want to employ any of these modes of exocytosis, STEPS supports modeling of full-collapse fusion, fusion where the vesicle is preserved (that may be used to model kiss-and-run or open and closed fusion), and partial release, with a user able to employ any of these different modes within the same model.

Within a full-collapse fusion exocytosis event in STEPS all luminal molecules are released into the extracellular space (Fig. 1(xii)) and all vesicle surface-bound species are transferred to the cell membrane (Fig. 1(xiii)). When fusion is not full-collapse and the vesicle is preserved vesicles can partially preserve luminal composition[55], and so in STEPS all or partial luminal contents are released (Fig. 1(xii)) at the modeler's discretion. As mentioned, fusion is also possible on internal structures (Fig. 1(xi)), enabling modeling of cycles such as the the post-synaptic AMPAR cycle by which AMPAR-containing vesicles fuse to endosome[4].

Figure 4a provides a basic validation of exocytosis in STEPS within the SSA framework, with further details of this model given in Supplementary Note 6. Figure 4e visualizes a full-collapse fusion event in STEPS of one synaptic vesicle in the active zone, where release of neurotransmitter into the extracellular space and retention of the SNARE complex and vesicle surface-bound species in the cell membrane can be seen. Supplementary Video 2 shows multiple exocytosis events in the same active zone model.

Endocytosis (Fig. 1(vi)) from an invaginated region of a cell membrane, and budding from internal structures such as liposomes[59] (Fig. 1(xi)) both involve the creation of vesicles, and are modeled in STEPS as events within the SSA. These events can optionally be modeled with a species dependency, meaning that the event is only active and available for application within the SSA if a certain species signature is met, for example if a certain number of clathrin molecules have been recruited to the region[60] defined as an `Endocytic Zone` within STEPS (Fig. 1(xiv)). This region may be defined as a collection of surface triangles in the tetrahedral mesh, or as molecular

population of a `Raft` (Rafts are described later in section Membrane Rafts: Formation, diffusion and interaction with their environment). Figure 4b–d provides a basic validation of endocytosis in STEPS within the SSA framework when the endocytosis region is either based on rafts (Fig. 4b) or comprised of mesh surface triangles (Fig. 4c).

STEPS is capable of modeling the complex molecular machinery that generates endocytosis events, but does not yet support adaptive meshes and so the geometric changes are not fully modeled. Instead, when an endocytosis event is chosen to take place, the spherical space just inside the endocytosis region is reserved and a vesicle is formed in place. This vesicle will contain the molecular makeup of the zone in which endocytosis occurred in its membrane, to mimic the molecular biology (Fig. 1(vi)). Figure 4d validates the expected production of vesicles during repeated endocytosis events. As dependent species are consumed, the rate of production of vesicles can be seen to decrease in this validation model. The endocytosis validation models are described in more detail in Supplementary Note 7.

### Vesicle surface protein and vesicle internal molecule transport, and interaction with environment

Vesicles contain proteins embedded in their membrane which allow them to interact with their environment through protein-protein or other interactions (Fig. 1(vii, viii)), or even interact with each other (Fig. 1(iii)). An important example of such interactions are those that allow synaptic vesicles to go through the process of docking and priming, which is controlled by interactions between vesicle proteins and proteins embedded in the cell membrane[61] (Fig. 1(viii)). Further, primed vesicles then sense calcium released into the cytosol to undergo the process of fusion and neurotransmitter release through a cascade of chemical interactions[53,62].

STEPS supports all of these situations by addition of a new type of reaction termed a `Vesicle Surface Reaction` by which reactants may exist in a number of regions: they may be compartment-based (e.g. cytosolic) molecules, molecules in the vesicle surface, or molecules embedded in the cell membrane or other membranes. A reaction between a molecule in the vesicle surface and the cell membrane can be used to model docking (Fig. 1(viii)), for example. In addition, reaction products may appear internally within the vesicle to model e.g. neurotransmitter pumps and filling (Fig. 1(vii)). These reactions are then solved stochastically as regular SSA reactions. This allows modeling of the rich interactions that vesicles may undergo with their environment and allows, for example, the entire synaptic vesicle cycle to be modeled to full molecular detail[35]. To faithfully model docking reactions specifically, optionally a maximum distance may be defined so that the reaction will only occur within a specified distance of the membrane surface, and vesicles may be immobilized when effectively docked close to the membrane (Fig. 1(ii)). The reverse reaction can also be modeled where vesicles may become undocked and mobilized, which is important if modeling kiss-and-run or open and closed fusion.

Figure 5 demonstrates our validation of the vesicle surface reactions by way of interaction with vesicle surface molecules and cytosolic species. We extend our regular validations[19] for these interactions, and demonstrate very close agreement in four different models, namely first order irreversible (Fig. 5a) and reversible (Fig. 5b) reactions, and second-order irreversible (Fig. 5c) and reversible (Fig. 5d) reactions, in every model demonstrating very close agreement to the expected solution analytically. Supplementary Note 8 contains further details of the model setup and parameters used.

**Vesicle interactions with each other.** A known behavior of vesicles is their ability to form functional clusters or pools[38,63] in which mobility is strongly reduced[64]. It has been proposed that such clusters are formed by vesicles tethering to each other for example in the case of synaptic vesicles by synapsin interactions[50,65]. STEPS models this phenomenon by allowing vesicles to loosely bind to each other through interactions between their surface proteins. A special type of reaction termed a `Vesicle Binding` event models this, alongside the reverse process `Vesicle Unbinding` (Fig. 1(iii)).

As vesicles bind in STEPS they form a computationally distinct complex termed a `Link Species` on their surface (Fig. 5f). Link Species have the property of length, which is by a user-defined upper and lower boundary, and the Vesicle Binding event can only occur if and when the resulting Link Species formed would be within its permissible bounds. Link Species allow some mobility of linked vesicles as long as any movement results in the Link Species remaining within its bounds. In addition, Link Species can undergo all the usual chemical interactions such as interaction with cytosolic or vesicle-bound molecules, and diffusion on the vesicle surface. Through this behavior, vesicles are therefore able to form clusters in which they exhibit loose, partial mobility, phenomenologically similar to their low mobility in pools[64]. Thus vesicle clustering and pool formation such as the classical reserve pool, recycling pool and readily releasable pool[38] can be faithfully modeled in STEPS, along with their interactions such as activation by calcium, and the process can be modeled to high molecular detail.

We provide a validation model of the Vesicle Binding reaction in STEPS, described in Supplementary Note 9, which has a known analytical solution to which STEPS output can be compared. Figure 5e shows close agreement between the STEPS model and analytically expected behavior of this model. In addition, Fig. 5f demonstrates cluster formation in which synapsin dimers bind vesicles to each other, showing liquid phase behavior[37].

In addition, Supplementary Video 3 shows a cluster model in which vesicles are bound to each other by synapsin dimers yet still exhibit some mobility. The video shows 300 vesicles that are bound together and 10 inert vesicles that do not interact with the cluster but can be seen to mix with it, demonstrating liquid phase properties of the cluster.

### Membrane rafts: formation, diffusion and interaction with their environment

Nanoscale domains of clustered proteins, lipids and other molecules in cell membranes play many important roles in signaling and trafficking[66]. Referred to as Lipid Rafts, these important membrane subdomains span a large range of sizes, reported as 10 nm to 200 nm in a definition that includes caveola[67], although a recent fluorescence resonance energy transfer study measured a lower bound of 5 nm[68]. Lipid rafts are dynamic in their molecular composition, and may be mobile and diffuse freely in the membrane or become bound to cytoskeleton and immobile[69]. Somewhat analogously to vesicles within cytosol and other cell compartments, they cannot be modeled as regular volumeless molecules within SSA methods and require special treatment.

We implement `Rafts` in STEPS as regions that occupy an exclusive, fixed radius within cell membranes. To mimic the biology, rafts may be mobile by a user-defined diffusion rate within cell surfaces (Fig. 1(xiii)) or immobile. They can undergo all the usual chemical interactions with their environment including with cytoplasmic molecules and surface-bound molecules and, in addition, rafts can be created upon exocytosis of a vesicle (as proposed for example in ref. 51) or form an endocytic zone from which a vesicle will be produced, sharing the surface molecular composition with the exocytosed or endocytosed vesicle (Fig. 1(vi, xiii)).

Since rafts are associated with a triangular meshed surface in a STEPS simulation, we apply the same diffusion rule for rafts as for regular SSA surface species. Figure 6a validates this implementation in a diffusion model where crowding effects are minimal, although it should be noted that diffusion rates can be affected by crowding if raft density is high. In addition, and similarly to our validation of vesicle surface reactions, we validate our implementation of raft surface reactions by models of first-order irreversible (Fig. 6b) and reversible (Fig. 6c) reactions, second-order irreversible (Fig. 6e) and reversible (Fig. 6f) reactions. As a useful modeling feature, it is also possible in STEPS to model formation of rafts when a particular species signature is met, that is when a user-defined list of species are present at a given number within a triangle. Conversely, dissociation and dispersion can be modeled if a raft molecule composition goes below a certain threshold. Figure 6d validates these implementations in a simple model that includes

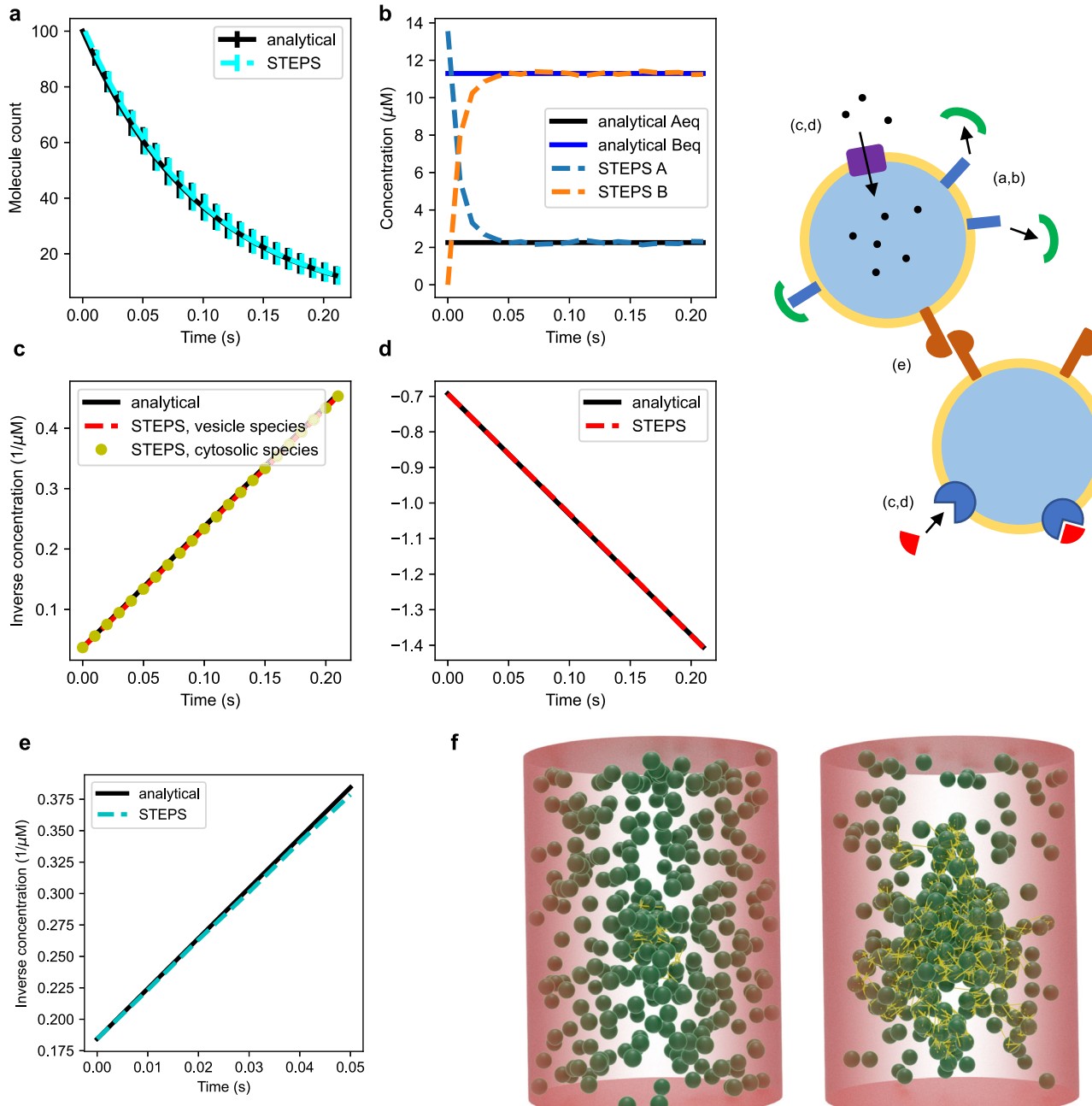

**Fig. 5 | Validation of vesicle surface molecule interactions.** As described further in text and Supplementary Note 8, **a** (*n* = 1000 independent simulations) and **b** validate first order surface interactions whereas **c** and **d** validate second-order interactions such as interactions with cytosolic molecules. **e** Validates vesicle binding interaction (vesicles binding to each other) which can be visualized in **f** as Blender output of a clustering model showing various stages of dynamic cluster formation (the synapsin dimer proteins that bind vesicles to each other are shown in yellow). The cluster can form or disperse based on cellular activity in the model.

both of these phenomena. Further details of all of these models can be found in Supplementary Notes 10 and 11.

### Computational performance in parallel
STEPS implements a parallel splitting algorithm for the vesicle solver, since an original serial implementation was found to be very limited in applications due to all the extra computation involved in supporting vesicle and raft modeling. Described in further detail in MPI implementation, we achieved an approximately 50X speedup on a realistic and detailed synaptic bouton model of Gallimore et al.[35], simulating 1 ms of biological time in 7.3 s of wall-clock time (averaged over a total run of 2 s biological time). This performance is sufficient to bring biological systems that operate on timescales of seconds or minutes such as the synaptic vesicle cycle into scope. If computer

cluster runtime constraints are a concern, checkpointing further extends the biological time that is achievable. Further improvements to MPI performance may be possible by parallelizing the vesicle routines and with further optimization to the MPI communication schemes in the future.

### Discussion
In this study we presented a new modeling tool that supports realistic simulation of vesicles and their many important biological functions such as uptake, transport and secretion, as well as lipid rafts. To our knowledge, this is the first approach to successfully combine such modeling in a hybrid framework with spatial SSA processes, enabling detailed chemical modeling of important biological systems such as the synaptic vesicle cycle. Every stage of development was carefully tested and validated, with many of the

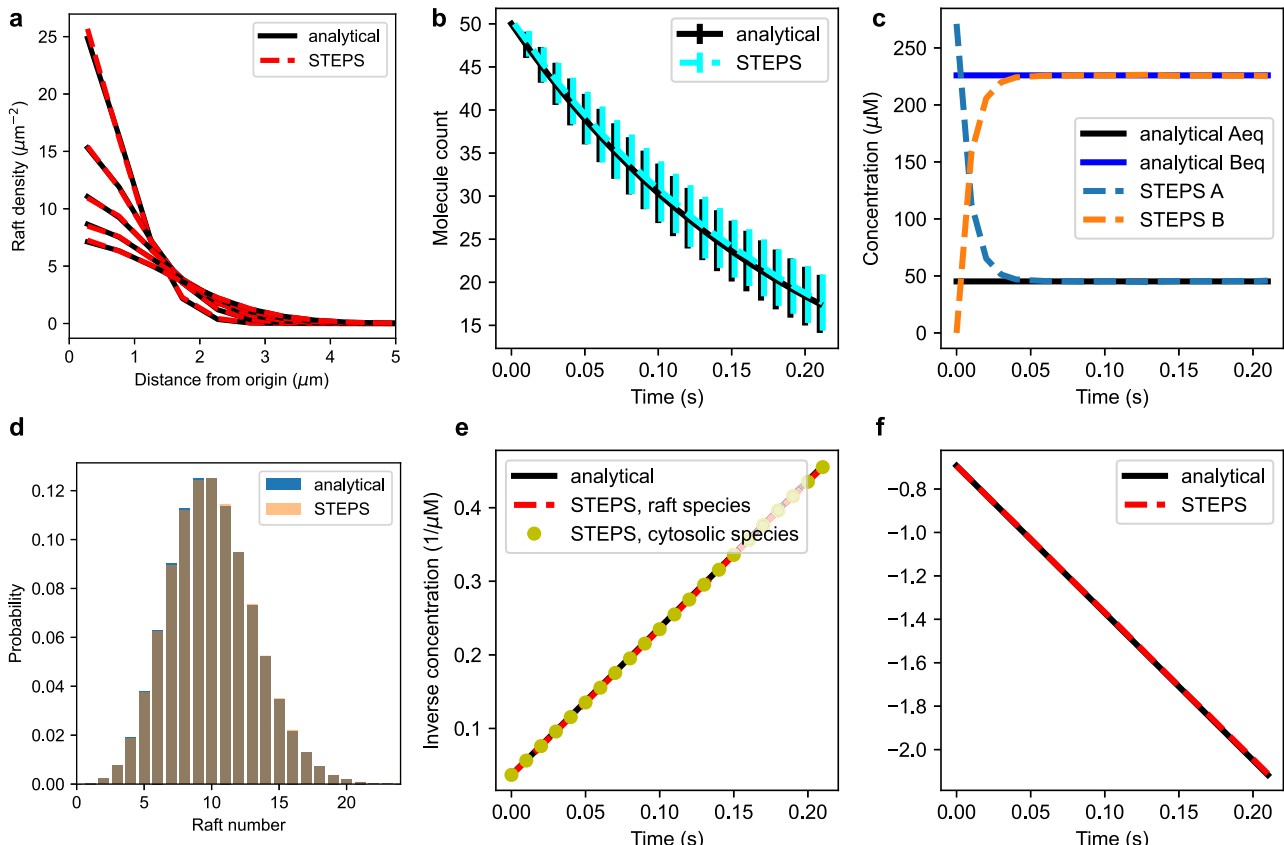

**Fig. 6 | Validation of rafts and their interactions. a** Raft diffusion on membranes under no-crowding conditions. **b** First-order irreversible (*n* = 500 independent simulations) and **c** first-order reversible interactions. **d** Combined raft generation and raft dissociation interaction model validates the SSA implementation of these phenomena. **e** Second order irreversible interactions and **f** second-order reversible interactions with membrane-bound molecules and cytosolic molecules. All models are compared to analytical solutions.

validations described in this paper, and implemented in realistic biological scenarios. In one example described by Gallimore at al.[35] and shown in MPI implementation, a detailed pre-synaptic bouton model simulated the synaptic vesicle cycle to full molecular detail.

The software was implemented in an MPI framework that took into account the unique challenges of modeling volumetric objects within the partitioned spatial SSA framework. Although this is a difficult system to parallelize, by taking advantage of the fact that vesicles diffuse much slower than smaller molecules we were able to sufficiently improve performance by orders of magnitude compared to a serial implementation, putting into reach biological systems operating on timescales of seconds or minutes. Such timescales are unreachable in molecular dynamic approaches.

We have developed this as a general tool that has the flexibility to enable application to a variety of modeling systems with realistic modeling of vesicles and lipid rafts and with the power to reach size and time scales of interest. The software is free for anyone to use under the GNU General Public License v3.

## Methods
### The new entities and kinetic processes in STEPS 5.0, and the relationship between them
STEPS follows an object-oriented programming style, and many new computational objects have been added in 5.0 for vesicle modeling, including Vesicles, Rafts, and their modes of interaction with their environment such as Vesicle Surface Reactions, Exocytosis and so on, as described in Results. Figure 7 summarizes the relationship between these objects with a color-coded description, and Supplementary Table 1 shows this information in tabulated form. One important feature is that

molecules on a vesicle surface (both Species and Link Species) have an exact position so that they can both undergo diffusion on the surface of the vesicle (Vesicle surface molecule diffusion), and so that their position can be found within the mesh for availability to the SSA (for example for a Vesicle Surface reaction or Vesicle Binding event). Since there is no concept of diffusion on a Raft surface in STEPS, molecules contained in Rafts do not have an exact position but are randomly assigned to Raft overlap triangles for availability to SSA processes (such as Raft Surface reactions).

### Vesicle definition and diffusion
Vesicles are approximated in STEPS as spherical volumes as obviously the most natural choice of the available geometrical primitives. This volumetric modeling of vesicles is a clear difference to the effectively point-particle SSA species in STEPS used to model smaller molecules such as ions, ion-channels and enzymes. Cellular vesicles vary in size; for example in neurons synaptic vesicles are approximately 40 nm in diameter and can transport a relatively small amount of neurotransmitter and surface proteins but can release quickly, whereas dense-core vesicles that transport neuropeptides are larger at 80 nm to 200 nm and operate more slowly[70]. A STEPS user specifies vesicle diameter that vesicles of this type will then assume during the simulation as effectively a sphere that operates within the realms of the tetrahedral mesh environment. A STEPS user is not limited in how many different types of vesicles they may define and so, for example, clear synaptic vesicles and dense core vesicles can both be modeled within the same simulation with differing properties, such as size, rate of diffusion, fusion time course, membrane protein composition, and so on. The user may then also define a fixed timestep on which to update vesicle positions by diffusion, although this will default to 1 ms if not specified.

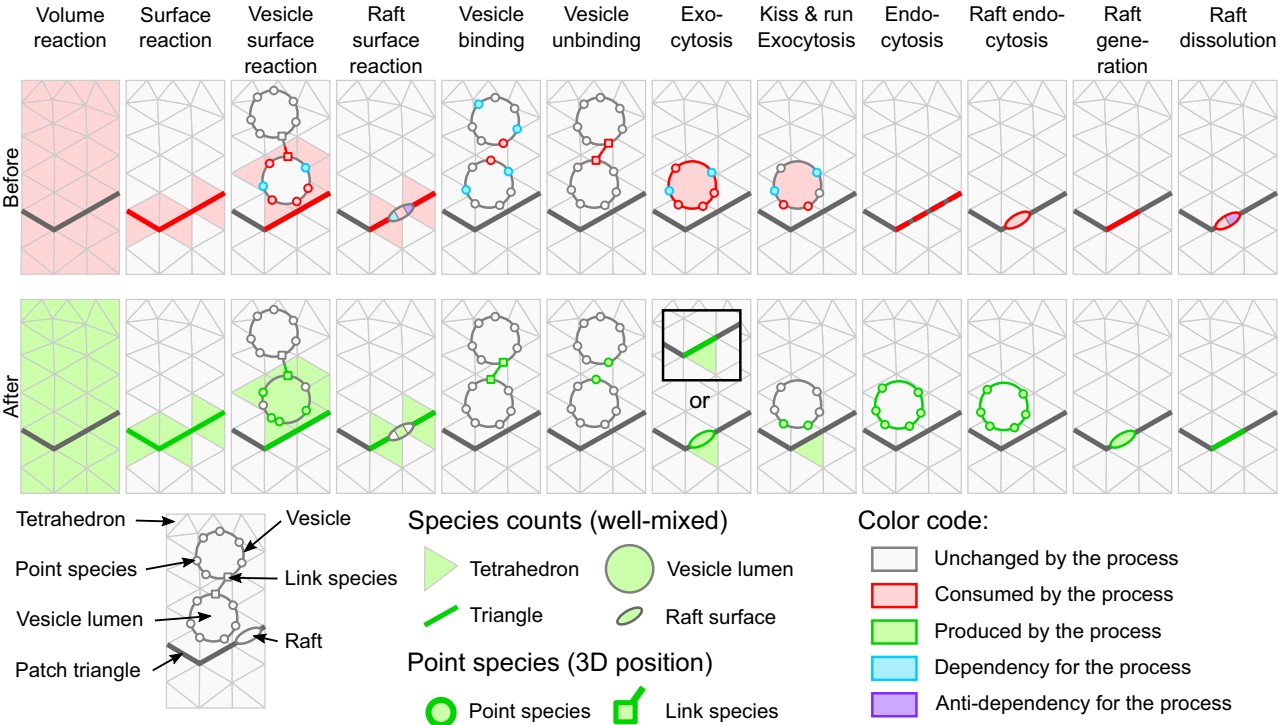

**Fig. 7 | Schematic summary of kinetic processes.** Each kinetic process (columns) is schematically explained by representing entities involved in it along with their states before (first row) and after (second row) the kinetic process happens. The type of entities involved is represented by different shapes, as indicated on the bottom left part of the figure. The role that the entities play in the kinetic process is indicated by their color (see bottom right legend). Species can either have a specific 3D position (point species and link species) or be held as a count in a well-mixed container (species in tetrahedrons, triangles, vesicle lumen, and raft surface). In addition, vesicles also have a specific 3D position and rafts are centered on specific triangles. Schematics on the top row (before the kinetic process happens) indicate in red which entities can be consumed by the process, in blue which species can be a dependency for the process (the process cannot happen unless these dependencies are met) and in

purple species that are an anti-dependency for the process (the process can be prevented from happening by the presence of species). When dependencies are not met, or when anti-dependencies are met, the kinetic process has rate 0 in the SSA and thus cannot happen. Different colors on rafts indicate the different roles that species on the surface can take. Schematics on the bottom row (after the kinetic process happens) indicate in green the entities that are produced or modified by the process. Exocytosis can either release the species on the surface of the vesicle in a patch triangle or as a newly created raft. Endocytosis happens on an endocytic zone which consists in a set of patch triangles (represented by the red dashed line); all the species in these triangles are moved to the surface of the newly created vesicle. A more detailed summary of the kinetic processes can be found in Supplementary Table 1.

In order to model Brownian motion on these regular timesteps, $\Delta t$, a diffusing vesicle's next position is chosen by sampling the x, y, z displacement separately:

$$
\begin{aligned}
\delta x &= r1 * \sqrt{2D\Delta t} \\
\delta y &= r2 * \sqrt{2D\Delta t} \\
\delta z &= r3 * \sqrt{2D\Delta t}
\end{aligned}
$$

where r1, r2, r3 are random numbers generated on the standard normal distribution. The new position is then tested to see if it is a valid position or not. For example, the vesicle must not overlap any other vesicle nor must it cross a boundary such as the mesh surface. To optimize the search of other vesicles, only vesicles known to overlap the tested tetrahedrons are checked. To calculate whether a vesicle crossed a compartmental boundary requires knowledge of the sphere-tetrahedron overlap of the new position (see Sphere-Tetrahedron overlap).

Figure 3a shows diffusion distances for a diffusion coefficient of mobile synaptic vesicles of 0.06 $\mu m^2 s^{-1}$ [8] for timesteps of 0.1 ms and 1 ms. Since diffusion distance is lower than vesicle diameter, meaning that the vesicle will continuously overlap some volume when diffusing, these appear to be reasonable choices for this type of vesicle, allowing a more optimal overlap search (see Sphere-Tetrahedron overlap). Sources of error, and reporting to user expands on when the timestep can be too large, and how this is reported to the user, with suggestions for a better value.

## Sphere-Tetrahedron overlap

STEPS uses an external library to determine the sphere-tetrahedron overlap[71] so that the set of tetrahedrons that a vesicle overlaps can be found. For application of this algorithm, it is essential to avoid testing all tetrahedrons in the entire mesh due to runtime concerns. If a vesicle is diffusing by a diffusion distance that is smaller than the vesicle's diameter, the set of tetrahedrons previously overlapped are taken as the starting point for the overlap search because there is guaranteed overlap in this set. If the diffusion distance is bigger than the vesicle diameter, a walking algorithm is used to find a path between the current vesicle center and the future vesicle position. The algorithm consists in a variant of the A* algorithm[72]: a set of tetrahedrons is initialized with the tetrahedron containing the current center of the vesicle, neighboring tetrahedrons are then explored by processing first tetrahedrons closest to the destination. The algorithm stops when the tetrahedron containing the destination has been found or when all tetrahedrons that are below a threshold distance $d_\theta = 3 \parallel \vec{p_s} - \vec{p_e} \parallel$ from the destination have been explored (with $\vec{p_s}$ the starting position and $\vec{p_e}$ the destination). With this distance threshold, this algorithm prevents vesicles from jumping over gaps in the mesh. When the vesicle diameter is small compared to the size of a tetrahedron, it is possible for $d_\theta$ to be too small and prevent exploration of even the immediate tetrahedron neighbors. To avoid this, the algorithm always explores at least 10 tetrahedrons before stopping.

A breadth-first search is then used to test overlap of tetrahedrons, stopping if no overlap can be found within a layer or overlap reaches 100% of the vesicle volume (which means that the new position is good). Overlap

under 100% means that the vesicle is not full covered by tetrahedrons and so the position being tested is outside of the permitted environment (i.e outside of compartmental boundaries).

## Vesicle-mesh overlap effects on molecular reaction and diffusion

To approximate the effect of vesicle presence on the mobility of other molecules (modeled within the operator-splitting framework[22]), we apply a correction to the finite-volume diffusion rate[19] of molecules proportional to the volume occupancy. The modified diffusion rate from tetrahedron $k$ to tetrahedron $l$ of species $i$ is:

$$d_{k,l}^* = \frac{D_i A_{k,l}}{V_k^* \delta x_{k,l}}$$

where $D_i$ is the macroscopic diffusion coefficient of species $i$, $A_{k,l}$ is the face area between tetrahedrons $k$ and $l$, $V_k^*$ is the reduced volume by vesicle occupancy, and $\delta x_{k,l}$ is the barycenter-barycenter distance. This reduced volume effect intuitively reduces the dwelltime for a molecule inside a tetrahedron proportional to the reduced volume, whilst also ensuring SSA species cannot occupy a fully overlapped tetrahedron and so do not occupy the same space as a vesicle. Note that mathematically the diffusion rate will approach infinity as the reduced tetrahedral volume approaches zero (meaning complete coverage by a vesicle) and is effectively replaced by a high rate for diffusion to a neighbor within the operator-splitting framework if this situation occurs. In realistic meshes, however, usually full coverage does not occur (Fig. 2c) (and won't occur unless tetrahedrons are very small, Fig. 2b) and in addition molecules are expected to evacuate tetrahedrons before full overlap occurs. Diffusion to a fully-overlapped tetrahedron is not permitted.

We tested the effect on regular diffusing SSA species in a model with 10% volume occupancy by vesicles, as described in Supplementary Note 12. In all cases, the tetrahedrons captured the binomial distribution of molecules with low error, as shown in Supplementary Fig. 4. The error in the fit to the binomial distribution was always less than 0.5%. This means that no spatial bias is introduced on the SSA species distributions by the modified diffusion rates.

STEPS also applies a volume correction to SSA reaction rates in order to capture effects of a reduced volume. For example, the rate, $c$, of a particular second-order reaction taking place in tetrahedron $j$ is:

$$c^* = \frac{K}{N_A V_j^*}$$

where $K$ is the macroscopic reaction rate, $N_A$ is Avogadro's number and $V_k^*$ is the reduced volume by vesicle overlap. In this sense, the volume that vesicles occupy is not regarded as part of the well-mixed volume of a tetrahedron. This can be expected to accentuate small-volume errors on reaction rates[73] if overlap is high. However, in practice, because overlap is transient and often relatively small, this effect also may be small or negligible because the effective volume is still usually within the accurate size window[19]. It should be noted that all validations presented in this paper, such as the vesicle surface molecule interactions presented in Vesicle surface protein and vesicle internal molecule transport, and interaction with environment and Vesicle interactions with each other pass within this framework. Unimolecular reactions are unaffected. The following section describes under which circumstances the second order reaction error can become large, how this is reported to the user, and steps that can be taken to reduce the error.

## Sources of error, and reporting to user

In this section we describe sources of error that can appear in models, often either by model design or from mesh construction, and how STEPS reports these expected errors and suggests modifications to the model in order to reduce or nullify these errors.

**Basic checks on STEPS models**. Firstly, when a STEPS simulation is created, a series of automatic checks are run on the model parameters to flag any potential mistakes or problems, with warning messages displayed to the user. For example, if any molecular species only appear on the right-hand side of reactions then a warning is displayed because nothing in the model consumes the species, and reaction rates are compared to other reactions in the model of similar type to check for any obvious outliers that could be simple typing errors. We have extended these checks for the new reactions that take place in STEPS 5.0. In addition, basic checks specific to vesicle and raft modeling have been added. STEPS checks whether the sizes of vesicles and rafts are larger than the size of the mesh and whether diffusion coefficient for vesicle surface diffusion are so high that one vesicle $\Delta t$ would result in a random position on the surface, since these are indicative of modeling errors.

We now describe more detailed checks on diffusion and reaction rates, and when warnings and suggestions are displayed to the user.

**Vesicle diffusion**. Perhaps the most obvious source of error when modeling vesicle diffusion on fixed time steps, $\Delta t$, is if the usual diffusion distance is too large. We suggest that this is the case when a high proportion of diffusion steps result in a vesicle diffusing greater than its diameter plus the smallest diameter of other vesicles in the model. A diffusion step of this distance will, in theory, allow a vesicle to pass through other vesicles whereas they would meet these obstacles if diffusing with a lower distance, and therefore a large diffusion distance could underestimate effects of crowding. Figure 8a shows, in the case where all vesicles in the system are of the same diameter, the proportion of diffusion steps over 2 diameters, as a function of the root mean squared diffusion distance. The fraction of vesicle diffusion steps above a distance $d$ can be obtained from $1 - F(\frac{3d^2}{6D\Delta t}; 3)$ with $F(x; k)$ the cumulative distribution function of the chi-square distribution with $k$ degrees of freedom and $D$ the vesicle diffusion coefficient. STEPS checks the model parameters and reports to the user if 5% of the diffusion steps are expected to be over a vesicle's own diameter plus the smallest diameter of other vesicles in the system, and suggests a vesicle $\Delta t$ to the user at which this ceases to be the case.

**Bimolecular reactions**. As previously noted, and described elsewhere[19,73], tetrahedron size can influence the simulated reaction rate of bimolecular reactions where molecules must diffuse to meet each other before reacting. Although, for realistic parameters, a voxel size window usually exists where relative error is low[19,23], the error can become large if tetrahedron sizes go below a certain size threshold- an important consideration for mesh construction.

The new components in STEPS 5.0 that support bimolecular reactions modeling and so are subject to these tetrahedron size considerations are:

- Vesicle Surface reaction (bimolecular)
- Raft Surface reaction (bimolecular)
- Vesicle Binding

We investigate this effect with a Vesicle Surface Reaction model where reactions take place between reactant molecules in the cytosol and on the vesicle surface. In Fig. 8b we demonstrate the error in our simulated reaction rate of the vesicle surface reaction validation model (Vesicle surface protein and vesicle internal molecule transport and interaction with environment), but with different meshes representing the same morphology of a 0.5 μm diameter sphere. 8 meshes were used with average tetrahedron size ranging from 120 nm (291 tetrahedrons) to 12.8 nm (265,307 tetrahedrons) (these two particular meshes can be seen in Fig. 2a, b). This spans approximately an order of magnitude in size, and therefore 3 orders of magnitude in volume. We simulate two cases, one where the vesicles are diffusing at 0.1 μm²s⁻¹ and one where vesicles are immobile, which could represent docked or clustered vesicles. We observe that the error generally stays low until 12 nm, although the error

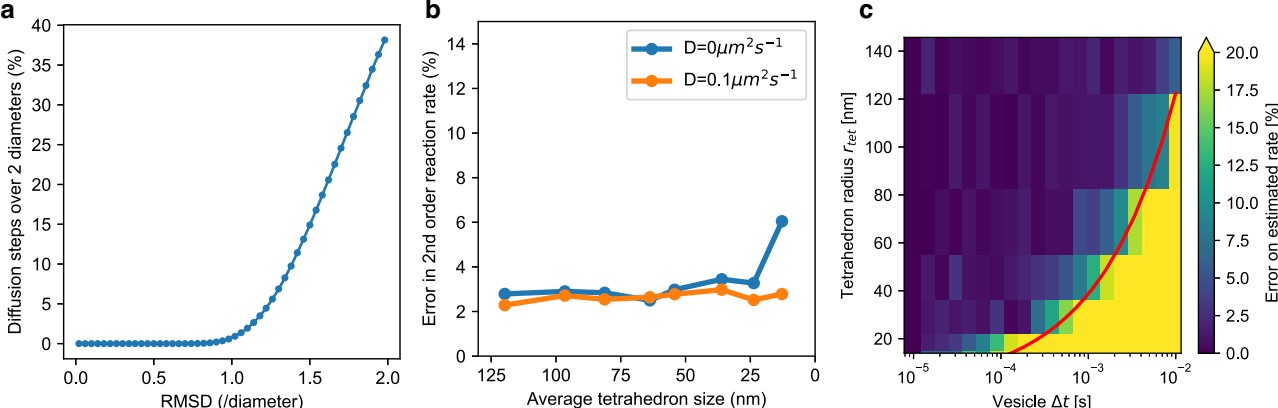

**Fig. 8 | Sources of error in vesicle models in STEPS. a** Vesicle diffusion: The percentage of diffusion steps that are over 2 vesicle diameters in length when the root mean square diffusion distance ranges from 0 to 2 vesicle diameters. **b** Bimolecular vesicle surface reactions with mobile cytosolic reactants: For different meshes representing the same 0.5 μm diameter spherical geometry, ranging from an average tetrahedron size of 120 nm (291 tetrahedrons) to 12.8 nm (265,307 tetrahedrons), the error in captured reaction rate is shown, when vesicles are stationary (blue line) or diffusing at 0.1 μm²s⁻¹ (orange line). The error is quantified by calculating the simulated rate by Supplementary Equation 6 and comparing to the expected rate

(n = 1000 independent simulations for each mesh) **c** Error on the rate of second-order vesicle surface reactions when a cytosolic reactant is immobile. Since reactants are immobile and vesicles diffuse relatively slowly, one cannot use the integrated rate law as ground truth to estimate the error; instead, we consider the simulations at the smallest vesicle $\Delta t = 10$ μs as ground truth for the expected rate. In these simulations, vesicles diffuse at 0.1 μm²s⁻¹. The red line depicts the threshold on vesicle $\Delta t$ from Supplementary Eq. (19). For each pair of average tetrahedron radius and vesicle $\Delta t$, the color in the heatmap denotes the error on the reaction rate estimated from the average concentration of reactants across n=20 independent simulations.

is usually slightly higher with immobile vesicles until this point. At 12 nm the error in the reaction rate with immobile vesicles increases sharply to 6.05% whereas it remains low with the diffusing vesicles, at 2.78%. This is because with immobile vesicles, the relative diffusion rate between the reactants is reduced, amplifying the small volume effect. Vesicle modeling can contain immobile or low-mobility vesicles by effects such as docking or clustering, and we conclude that it may be more difficult to capture bimolecular reaction rates under these scenarios with very small tetrahedrons. For these reasons, and for the observations in Fig. 8b and our previous observations[19], as well as observing the typical sizes of tetrahedrons in a realistic biological mesh as shown in Fig. 2c, STEPS checks a mesh and reports to the user if 10 percent of tetrahedrons are below a size of 20 nm. This check also captures simple mesh scaling errors.

Since we observed that the vesicle diffusion rate can influence bimolecular reaction rates, we can expect that the vesicle diffusion timestep may also have an effect. We investigated this under the scenario of immobile cytosolic reactants as a tractable mathematical problem and effectively worst case modeling scenario, where the observed error is smaller with mobile cytosolic reactants (see Supplementary Fig. 5 for cytosolic reactants diffusing at 1 μm²s⁻¹). Under this scenario, bimolecular reaction rates can be underestimated if the vesicle $\Delta t$ is too high. For higher vesicle $\Delta t$, vesicle surface species spend a longer time in each tetrahedron than would be expected if vesicle diffusion was continuous, immobile reactants get more depleted than they should and this can lead to an underestimation of bimolecular reaction rates. Figure 8c shows the error on the estimated rate as a function of vesicle $\Delta t$ and tetrahedron sizes. Based on the sizes of tetrahedrons in the mesh, STEPS computes a threshold value for vesicle $\Delta t$ (red line) above which the error on reaction rate is deemed too high, and displays a warning to the user, suggesting a more appropriate vesicle $\Delta t$. This threshold is computed with $\Delta t_\theta = \frac{r_{tet}^2}{15D}$ which corresponds to the average time needed for a point diffusing in 3D with diffusion coefficient $D$ to exit a sphere of radius $r_{tet}$ when its initial position is uniformly distributed in the sphere. For any given mesh $r_{tet}$ is taken so that the volume of the sphere is equal to the average volume of a tetrahedron. Details related to how the threshold was derived are available in Supplementary Note 13 and Supplementary Eqs. (10)–(19).

We should note that some reactions are not affected by the tetrahedron size consideration because they do not involve two reactant molecules. In STEPS 5.0 these are:

- Vesicle Surface reaction (unimolecular)
- Raft Surface reaction (unimolecular)
- Vesicle Unbinding
- Exocytosis
- Endocytosis
- Raft endocytosis
- Raft generation
- Raft dissolution

As an example, Supplementary Fig. 3 shows how a unimolecular vesicle surface reaction is not affected by tetrahedron size.

**MPI implementation**

We previously parallelized the reaction-diffusion and voltage solution in STEPS[21], yet the complex additional vesicle-related functionality requires a customized solution that builds on this scheme. Our approach is to implement a splitting method whereby rank 0 (which we term the VesRaft rank) carries out vesicle-related computations such as vesicle and raft diffusion, diffusion of vesicle surface species, and application of endocytosis and exocytosis (Fig. 9a, b), and the other available processes (which we term the RDEF ranks) carry out the regular SSA calculations based on our previous method, which now however include many new phenomena such as reactions on the vesicle and raft surfaces, fusion and so on (Fig. 9a, b). This is achieved by effectively partitioning vesicles and rafts on the RDEF cores into objects that we term vesicle and raft proxies (Fig. 9a). This is essential to maintain the partitioned-SSA method that is known to scale well in realistic reaction-diffusion models[21], with many vesicle-related phenomena operating within the SSA in parallel. Of course this requires frequent communication between the VesRaft core and the RDEF cores (Fig. 9b).

The new solver inherits the MPI communication scheme for reaction-diffusion and the voltage solutions in the previous operator-splitting implementation[21], while adding communications of complex data structures between the VesRaft rank and the RDEF ranks (Fig. 9b). These two types of solver contain different local data, thus extra synchronization in data inquiry functions are also needed. These new challenges are addressed by the implementation of customized MPI datatype encapsulation and conditional MPI operation templates. To simplify and reduce the usage of MPI communication of multiple data

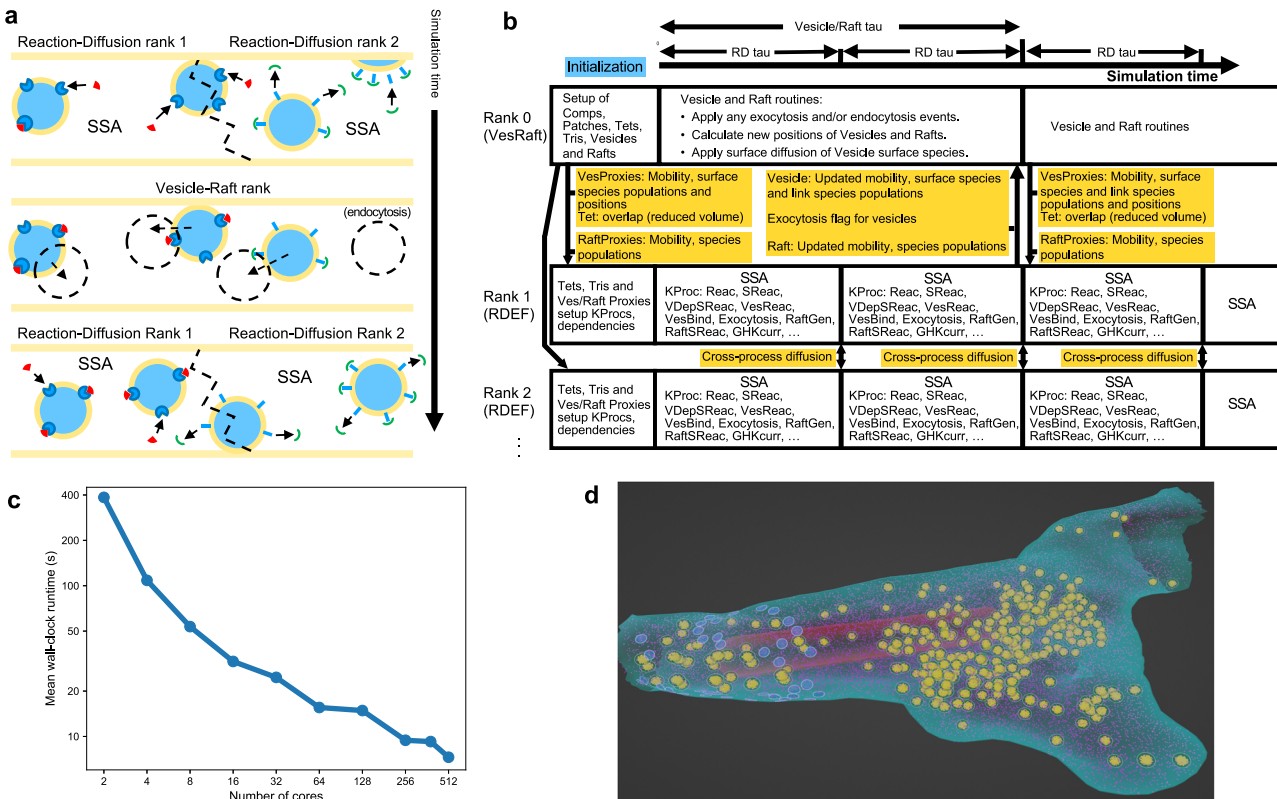

**Fig. 9 | The MPI implementation and Blender extension module. a** Schematic of the MPI splitting solution, in which n-1 `RDEF` ranks carry out reaction-diffusion and voltage calculations on partitioned mesh space, and on a slower clock the `VesRaft` rank (rank 0) calculates vesicle and raft diffusion and some other vesicle-related processes over all space, communicating new positions to the `RDEF` ranks (Note: for clear illustration purposes and to show how vesicles can change partition during simulation, a relatively large diffusion step of approximately one diameter is shown, although vesicles will often diffuse shorter distances in real simulations). **b** Detailed description of the parallel algorithm timeline. **c** Example parallel performance of a realistic synaptic bouton model of Gallimore et al.[35]. **d** Screenshot of the Blender extension model applied to the same synaptic bouton model.

entries in simulation objects (vesicles for example), data entries in object classes are firstly defined and encapsulated as custom datatypes using MPI's user-defined datatype mechanism. Such encapsulation, together with our STL (Standard Template Library) container based MPI broadcasting and gathering templates, allows us to pack objects with multiple primitive data types as a single vector and synchronize using minimal MPI calls. This helps to reduce the latency due to frequent synchronization of small size data entries and improves the simulation efficiency. Data output in the VesRaft rank is often irrelevant to RDEF ranks, and vice versa, therefore global synchronization is not necessary, particularly for vesicle-related data. We allow the user to switch communication mode for data inquiry functions by setting two condition flags in the solver: first, whether global output synchronization should be enabled; and second, if global output synchronization is disabled, which rank should receive the actual output. Note that network communication may still occur if the acquired data is not available in the output rank solver. In practice, we enable global synchronization by default, but disable it and set the VesRaft rank as output rank for heavy vesicle manipulation routines. Internally, this functionality is accomplished by a series of conditional MPI communication and data reduction templates, which automatically identify the data availability and perform the necessary data synchronization according to the conditions.

Figure 9c shows performance of our MPI-based solution on the model of Gallimore et al.[35] (visualized in Fig. 9d) run for 2 s biological time on a computer cluster with varying numbers of cores. Simulation wall-clock time reduces from 386 s on 2 cores (the lowest number of cores possible in our implementation) to 7.3 s on 512 cores per 1 ms of biological time, an approximately 50X speedup.

## Visualization with Blender

With the release of STEPS 4.1, the possibility to save simulation data to the HDF5 format was added. This data can then be loaded into scientific data visualization software such as Paraview[74]. Although well suited to the visualization of purely mesh-based data, this approach was not ideal for the hybrid SSA-vesicle simulations described in this paper. We therefore developed a separate `stepsblender` python package that can read simulation and mesh data automatically saved to an HDF5 file and visualize it in the Blender 3D computer graphics software (http://www.blender.org). Simulations can be visualized interactively or rendered as movies directly from Blender. The visualization includes vesicles, rafts, link species, regular species, as well as compartments, patches, vesicle paths, and endocytic zones. Fig. 9d shows an example still shot from the model described by Gallimore et al.[35] and output from the Blender extension module is also shown in Figs. 3b, 4e and 5f.

This new python package contains a command line tool to import the data into Blender as well as python modules that can be imported in python scripts to further customize the appearance of model objects. To avoid memory issues in Blender, only the data corresponding to the currently visualized timestep is loaded into Blender. The data loading from HDF5 file is handled in a separate process, potentially running on a different machine, allowing the visualization of data from a computing cluster without downloading the full HDF5 files.

## Statistics and reproducibility

Detailed information of the simulations shown in this study is provided in Supplementary Notes 1–13, including how many times each simulation was repeated with different sequences of random numbers, for which the mean

results of these simulations are shown in Figs. 3–6 and 8. Where standard deviation is shown, this is calculated with Python package NumPy from the sample size indicated in figure legends. All results shown in this manuscript, apart from the parallel performance study (shown in Fig. 9c), may be reproduced by model scripts available at https://github.com/CNS-OIST/STEPS_Validation/tree/main/vesicles, which also includes instructions for how to run the simulations, collect data and plot the results. The script for the parallel performance study is available on request.

## Data availability

All data used in this study are available in Zenodo at: https://doi.org/10.5281/zenodo.10901994[75].

## Code availability

STEPS 5.0.1 is available from the STEPS public release Github repository at: https://github.com/CNS-OIST/STEPS/releases/tag/5.0.1[76].

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

## Acknowledgements

Research reported in this publication was supported by the Okinawa Institute of Science and Technology Graduate University (OIST), and by Kakenhi Grant-in-Aid for Scientific Research Project Number 20K06593. We thank Tristan Carel of the Blue Brain Project, École Polytechnique Fédérale de Lausanne for his contributions to the vesicle code.

## Author contributions

E.D.S. conceptualized and led the study. I.H. led software development. J.L. and W.C. contributed to software development and I.H., J.L. and W.C. did pre-release testing and debugging. I.H. and J.L. wrote the Python model scripts, ran simulations and collected data. J.L. wrote the Blender visualization module. A.R.G. and S.Y.N.S. designed features of the software and tested the implementations. I.H., J.L., W.C. and E.D.S. wrote the manuscript. All authors read and approved the final version.

## Competing interests

The authors declare no competing interests.
