## [Peer review file · Communications Biology]

Reviewers' comments:

Reviewer #1 (Remarks to the Author):

In this paper, the authors presented a general vesicle modeling tool STEPS that has been designed for wide application to a variety of cell models, implemented within voxel-based approach to modeling reaction-diffusion processes in realistic mesh reconstructions of cell tissue. I think this study is of a potential interest to the readers including biochemists, neurochemists, physiologists and mathematicians.

I have some questions regarding to this manuscript.

1. Abstract. Please give a full definition of STEPS.

2. Figure 1. Is there an order in the abbreviated symbols in the figure, such as i, iii, viii...? It seems hard to me to follow it.

3. About "in the range of 5% and 80% for the vesicles undergoing kiss-and-run as opposed to full exocytosis" in the part of "Exocytosis, endocytosis, fusion and budding", the authors cited a review published in 2006 instead of the original research paper. Regarding this part, there are a bunch of papers showing besides full and kiss-and-run, there is another mode called "open and close" or "partial release" reported recently, where a significant but not entire NT is released during exocytosis. See an example in *Angew. Chem. Int. Ed.* 2015, 54, 11978. I think including this mode is necessary for this review.

4. Can STEPS define the different types of vesicles, including transparent vesicles and dense-core vesicles?

Reviewer #2 (Remarks to the Author):

The manuscript by Hepburn et al. presents an extension of software for biological simulations. The authors have developed the software STEPS that is a voxel-based simulator for reaction-diffusion kinetics in cells and organelles. In this study, they added the creation and removal of vesicles. The vesicles are involved in many biological functions and transports of proteins and nutrients. The software can treat active ballistic motion and passive diffusion of vesicles and the diffusion of reactant on the vesicle surface. Also, it can handle the reaction-diffusion kinetics on vesicles. The software is parallelized by MPI. The authors tested several simple conditions to check the code by comparing them with analytical solutions.

The software is well developed to treat many different conditions. It is a nice tool for biological simulations. However, each method or technique is not new. Scientifically new results are not included. Therefore, I consider that this manuscript does not include a new biological insight for publication in *Communications Biology*. A specific journal on computational biology is more suitable.

Reviewer #3/#4 (Remarks to the Author):

% Hybrid vesicle and reaction-diffusion modeling with STEPS

This markdown text can be converted to html, latex, and many other formats:

```
```sh
```

```
pandoc -t [html5,latex,docx] -s -o review.[html,tex,docx] ThisText.md
```

```
```
```

Review

=====

The work behind the manuscript is monumental. The alternative, less mechanistic approach is to model the average effect of vesicle activity with e.g. ordinary differential equations. The authors hybrid approach treats vesicles with an unprecedented accuracy for the model

class (stochastic systems biology models).

A lot of care has been put into the development of the software:

- accuracy
- performance
- parallelization (division of tasks into MPI processes)
- validation of the used numerical methods.

All used modeling techniques were checked with simple setups with known analytical solutions.

The manuscript is well written, carefully organized and easy to read.

We have however some comments and suggestions on things that could be improved.

General comments

1. The discussion part or the supplement should show (more) where the trade offs are (if any), and also mention error sources:

- under which circumstances do any of the chosen methods fail?
- which reasonable errors can users make to obtain wrong results?
- how are unreasonable settings discovered/reported to the user?
- which approximations were discarded as too inaccurate during development?
- which simulation parameter settings result in too large approximation errors

2. Consider replacing all approximate values in the text with ranges, or numbers with error quantification (standard deviations of the mean).

3. It is not entirely clear to us how far vesicles move per time-step compared to a typical voxel edge-length. Please clarify.

4. Please make a graphic depicting the size difference of a typical tetrahedron and the vesicle, next to each other, with size information in the caption.

5. Please summarize somewhere (maybe in a table the supplementary) how the different biological entities are represented within your approach and how they interact. For a list of entities:

- intracellular molecules
- vesicle contents
- rafts
- vesicles
- vesicle surface proteins
- membrane bound proteins

which entities have a **position**, a **size**, a **count** within a **tetrahedron**, a count within a **triangle**. A term like point-particle implies a position, which this model class typically doesn't have except for the location within the mesh (or compartment id), so we suggest to make this very explicit.

If we have understood it correctly:

- The vesicles are exact spheres, with size and center location.
- The other intracellular molecules are counts within voxels, or triangles (when they are membrane bound).
- Molecules on vesicle surfaces maybe have a position, unclear.
- Membrane molecules that are not on vesicles, are counts

within triangles.

- Interactions between intracellular molecules are stochastic mass-action kinetics based within voxels (i.e. increase/decrease of counts)
- In contrast interactions between vesicle membrane molecules depend on the distance between vesicles and/or distance to neuron membrane.

A big table would make it easier to follow the slightly different handling of these entities within the entire modeling approach.

6. We have to guess how big the voxels are, judging from the depicted realistic models (blender visualization) the tetrahedra are very small, as the membranes have very detailed curvatures. This high polygon count implies that the tetrahedra are much smaller than the vesicles (roughly by an order of magnitude). If this is true (and it probably is not as this would contradict other statements in the text), then the partial covering of tetrahedra by vesicles is a vesicle edge effect, and thus doesn't effect very many voxels.

Within this **small voxel setting**: would it be reasonable (fast) or too inaccurate to approximate the partial tetrahedron volume covered by a vesicle to be proportional to the number of vertices inside the vesicle?

In any case please expand on the typical size of vesicles, tetrahedra, time-steps (in each context) in the form of a table or graphic.

7. Please clarify what a **species signature** is. We assume that it is a list of species that must be present on a vesicle's surface.

Specific comments/suggestions

8. ****Line 436****: regarding "obviously", please clarify this point a little:

- + what happens if the step size is too big?
- + will the contents of tetrahedrons not evacuate quickly enough once the vesicle moves on top of it? The rest of the manuscript implies that the contents are suddenly moved to neighboring (not covered) voxels, but this may not be obvious to everyone.
- + might the simulation break or become inaccurate with bad voxels sizes, where total covering happens a lot?

9. ****Fig. 6**** suggests that vesicles move their entire diameter in one step, that seems fast, and contradicts the previous statement that they should move 10 nm per step. If this is because the illustration works better this way, then maybe it needs some minor adjustments with gray **in-between** circles of vesicle positions.

10. ****Line 64****: include a citation/reference for **docking**, **priming**, and **fusion**.

11. ****Line 173****: "user defined speed" (here and at other places within this section and suppl. note 4 and 5), please clarify if the dwell time is included when you speak about speed or not.

12. ****Line 190****: please clarify the statement in parentheses. What step-size and stochastic effects are you referring to?

13. ****Line 370**** : these four combinations of **reversible** and **reaction order** could form a table

| | first order | second order |
|-----:|:-----:|:-----:|
| irreversible | Fig. 5b | Fig. 5e |
| reversible | Fig. 5c | Fig. 5f |

14. **Line 387** : Maybe split this long sentence into two;
suggestion: *We tested our initial MPI implementation in a realistic biological scenario. In this stringent test, we achieved a good reduction \[...\]*
15. **Line 437** : Include the symbols of the different quantities (diffusion coefficient, time step, diffusion distance) already here like this: "\[...\] for a diffusion coefficient D of $\sqrt{0.11} \times \sqrt{\text{square}\ \mu\text{metre}\ \text{per}\ \text{second}}$ \[...\]" or similar.
16. **Line 454**: "If a vesicle is diffusing \[...\]", please reformulate. Maybe with: *diffusion distance smaller than the vesicle's diameter*.
17. **Line 489**: "as shown in as shown in" is doubled.
18. **Line 438**: "0.11 $\mu\text{m}^2/\text{s}$ " and "10 nm" diffusion distance. Is this correct? Please double check. We get roughly 0.8 nm:

```
``R
D <- 0.11e-12 # m2/s
dt <- 1e-6 # s
dr <- sqrt(6*D*dt) # 8.124e-10 m
``
```

19. **Line 442**: perhaps, in one location early on use the term *Brownian motion* in connection with these equations. (can be a footnote)
20. **Line 459**: "\[...\] a walking algorithm \[...\]": this is a little unclear. We can imagine what it means, but perhaps we are mistaken: you start in a random or user specified location and make Gaussian updates to the current location until a valid location is found.

Comments regarding Supplement
=====

21. **Note 4 & 5**: Please mention *step sizes*, *dwell-times* (or possibly omission of), and state whether the *mean speed* takes dwell-time into account.
22. **Note 6**: Was the simulation environment a sphere with 2d-diffusion, where diffusion happens on the edges between triangles? Or was it a thin volume (like a peel). I.e. is it *literally* 2D or *effectively* 2D. Please state explicitly.
23. **Note 6**: Regarding *iterations*: please clarify whether these are time-step iterations, or repeated simulations with different random number seeds (a statistical ensemble)?
24. **Note 6**: What is naïve planar diffusion with angular displacement on a vesicle? Please either expand or omit entirely.
25. **Note 9**: Fig. 2b should be Figure 4b. In paragraph starting with "This model \[...\]", below equation (6), "reversible" contradicts the previous statement above equation (6) where it is labeled "irreversible".
26. **Note 10**: We suggest to change "because only one type of species was involved" to *because both reactants are of the same species, their initial concentrations are identical and we can use eq (6)* (get to the point about equal initial concentrations).

27. **Note 12**: suggest to remove the word "iterations" at the end to not confuse with discrete time-steps (which are iterations within one simulation). Generally, once you have iterations within iterations or models within models, more precise language can help a lot. But, perhaps these iterations *are* time-steps and we misunderstood.

Notes on spacing, typesetting, and LaTeX

These are suggestions, or very minor corrections, listed by line:

30

: Is the section title "Main" deliberate?

154

: Change the math ``\mu`` to ``\upmu`` for an upright μ .
Or, even better:

```
```\latex
\usepackage{siunitx}
...
\unit{\square\micro\metre\per\second}
\qty{36}{nm}
```\
```

using the ``siunitx`` package. Upright μ ``\upmu`` is from the ``upgreek`` package (on page 438 there is an upright μ , so the authors are probably well aware). The command ``qty`` from ``siunitx`` also puts a half-space between number and unit and allows the user to switch between different error notations (concise, or \pm). Same μ inconsistency in the caption for Fig. 2. In the same caption ``\mu`` is instead a latin ``u`` in `0.5um2/s`, which is sometimes permitted as an ASCII replacement for μ (``\qty{0.5}{\micro\metre^2/s}`` would work as well if you want to type less).

174

: ``\sim`` (\sim) should perhaps be ``\approx`` when approximate values are given. Perhaps even better: use ranges and error notation, so that it is clear to everyone how approximate those values are. On line 351 a range is given as " $\sim 50\text{nm}$ to hundreds of nanometers", here it could be better to write **tens/dozens/several to hundreds of nanometers** or **between approx. 50 nm and 1000 nm** (or whichever numbers seem reasonable). The cited publication, [63], does a very similar thing, so this is probably fine, but they don't write it in the same sentence.

174

: ``myosinV`` should be **Myosin V**, probably.

Authors' response

We thank the reviewers for their detailed review of our manuscript. Such a thorough review has, we believe, enabled us to make substantial improvements to the manuscript, and also to the code. We have taken all comments on board and tried to address all suggestions and concerns as fully as possible, and below we give a point by point response to all comments received. Since this has involved substantial investigations and even additions to the software (as we will outline in individual responses) this has taken some time, but we feel the improvements were worth this time and effort. Please see below for detailed responses.

Reviewer 1

Review

In this paper, the authors presented a general vesicle modeling tool STEPS that has been designed for wide application to a variety of cell models, implemented within voxel-based approach to modeling reaction-diffusion processes in realistic mesh reconstructions of cell tissue. I think this study is of a potential interest to the readers including biochemists, neurochemists, physiologists and mathematicians. I have some questions regarding to this manuscript.

Detailed comments

Referee comment	Author reply
1. Abstract. Please give a full definition of STEPS.	A full definition of STEPS has been added to the Abstract, and the rest of the Abstract has been modified slightly to stay within the 150 word limit.
2. Figure 1. Is there an order in the abbreviated symbols in the figure, such as i, iii, viii...? It seems hard to me to follow it.	We have reordered the symbols in Figure 1 to follow the order in which they first appear in main text.
3. About “in the range of 5% and 80% for the vesicles undergoing kiss-and-run as opposed to full exocytosis” in the part of “Exocytosis, endocytosis, fusion and budding”, the authors cited a review published in 2006 instead of the original research paper. Regarding this part, there are a bunch of papers showing besides full and kiss-and-run, there is another mode called “open and close” or “partial release” reported recently, where a significant but not entire NT is released during exocytosis. See an example in Angew. Chem. Int. Ed. 2015, 54, 11978. I think including this mode is necessary for this review.	We have rewritten this section based on the Reviewer’s comments. The figures of 5% to 80% are based on many different works to give this range, not one research article, so we have simply removed reference to these specific values as part of the rework of this section. We have added “open and close” and “partial release” to this brief review of this topic, and have added the suggested reference for partial release (lines 260-266). Article word constraints (here and in other parts of the paper) prevent us from going into any more depth on this topic, but we do cover the full ground of what STEPS is currently capable of supporting for exocytosis modeling (lines 266-279).
4. Can STEPS define the different types of vesicles, including transparent vesicles and dense-core vesicles?	Yes, these different types of vesicles can be modeled distinctly within the same simulation and with different properties. We have expanded this description in Discussion to make this clearer (lines 489-493).

Reviewer #3/#4

Review

The work behind the manuscript is monumental. The alternative, less mechanistic approach is to model the average effect of vesicle activity with e.g. ordinary differential equations. The authors hybrid approach treats vesicles with an unprecedented accuracy for the model class (stochastic systems biology models).

A lot of care has been put into the development of the software:

- accuracy
- performance
- parallelization (division of tasks into MPI processes)
- validation of the used numerical methods.

All used modeling techniques were checked with simple setups with known analytical solutions.

The manuscript is well written, carefully organized and easy to read.

We have however some comments and suggestions on things that could be improved.

General comments

Referee comment	Author reply
-----------------	--------------

1. The discussion part or the supplement should show (more) where the trade offs are (if any), and also mention error sources:  • under which circumstances do any of the chosen methods fail? • which reasonable errors can users make to obtain wrong results? • how are unreasonable settings discovered/reported to the user? • which approximations were discarded as too inaccurate during development? • which simulation parameter settings result in too large approximation errors 	Thank you for this comment, and we agree that these aspects were somewhat lacking in our initial submission. We have spent quite some time making significant changes not just to the manuscript but also to the software itself, based on these comments, investigating sources of error and improving reporting to users. We have also added a new Figure (Figure 8) and a new section in Discussion (Sources of error, and reporting to user. From line 585) summarizing this work. We focused on quantifying errors related to vesicle diffusion (from line 606) and bimolecular reactions (from line 624) involving chemical species on the surface of vesicles or rafts. We investigated under which conditions the vesicle Δt could be too high and lead to unrealistic vesicle jumps or errors in the rate of bimolecular reactions, and how the coarseness of the mesh could also lead to errors in the rate of bimolecular reactions. We then added automatic checks to STEPS so that problematic values for the vesicle Δt or the size of tetrahedrons are reported and adequate values are suggested to the user. In addition to these checks, we also added a series of more basic checks that can prevent typing or scaling mistakes in the sizes of vesicles, raft or the values of vesicle surface diffusion coefficients, which are described in lines 590-605.
2. Consider replacing all approximate values in the text with ranges, or numbers with error quantification (standard deviations of the mean).	We have removed all approximate values in text, with further details covered in point 30 below.
3. It is not entirely clear to us how far vesicles move per time-step compared to a typical voxel edge-length. Please clarify.	In addition to a new figure showing edge lengths over the range of meshes used in this study (see next point), we have also expanded on the description of how far vesicles move (where previously we only had one example approximate value given in Discussion). While this is of course dependent on model parameters i.e. the vesicle diffusion coefficient and time step (Δt), we give a realistic example in Figure 3a for synaptic vesicles with two typical values for vesicle Δt. We have added a sentence comparing this diffusion step to typical tetrahedron edge-lengths shown in Figure 2c (lines 170-172).
4. Please make a graphic depicting the size difference of a typical tetrahedron and the vesicle, next to each other, with size information in the caption.	We have added a new figure, now Figure 2, with this information. The figure covers the range of tetrahedron sizes used for our validation studies from largest (panel a) to smallest (panel b) to show the full range of modeling scenarios, at least within our study. We also include our most biologically realistic mesh in panel c to show a typical modeling problem. The binned tetrahedron sizes are shown in the top panels over all tetrahedrons in the meshes. For these examples, we compare to vesicles of diameter 40nm representing synaptic vesicles.

5. Please summarize somewhere (maybe in a table the supplementary) how the different biological entities are represented within your approach and how they interact. For a list of entities:

- intracellular molecules
- vesicle contents
- rafts
- vesicles
- vesicle surface proteins
- membrane bound proteins

which entities have a *position*, a *size*, a *count* within a *tetrahedron*, a count within a *triangle*. A term like point-particle implies a position, which this model class typically doesn't have except for the location within the mesh (or compartment id), so we suggest to make this very explicit.

If we have understood it correctly: - The vesicles are exact spheres, with size and center location. - The other intracellular molecules are counts within voxels, or triangles (when they are membrane bound). - Molecules on vesicle surfaces maybe have a position, unclear. - Membrane molecules that are not on vesicles, are counts within triangles. - Interactions between intracellular molecules are stochastic mass-action kinetics based within voxels (i.e. increase/decrease of counts) - In contrast interactions between vesicle membrane molecules depend on the distance between vesicles and/or distance to neuron membrane.

A big table would make it easier to follow the slightly different handling of these entities within the entire modeling approach.

6. We have to guess how big the voxels are, judging from the depicted realistic models (blender visualization) the tetrahedra are very small, as the membranes have very detailed curvatures. This high polygon count implies that the tetrahedra are much smaller than the vesicles (roughly by an order of magnitude). If this is true (and it probably is not as this would contradict other statements in the text), then the partial covering of tetrahedra by vesicles is a vesicle edge effect, and thus doesn't effect very many voxels.

Within this *small voxel setting*: would it be reasonable (fast) or too inaccurate to approximate the partial tetrahedron volume covered by a vesicle to be proportional to the number of vertices inside the vesicle?

In any case please expand on the typical size of vesicles, tetrahedra, time-steps (in each context) in the form of a table or graphic.

7. Please clarify what a *species signature* is. We assume that it is a list of species that must be present on a vesicle's surface.

Thank you for this comment. We have added both a table and a Figure, with supporting descriptions in text (**lines 461-476**) to get all of this information across clearly. Figure 7 shows the behaviour of the biological entities and the relationship between them in a graphic because we felt this was the clearest way to get this information across in the manuscript. We have also added this information in table form with Supplementary Table 1.

We did investigate this suggestion to see if we could indeed support such an approximation in the small voxel setting. We investigated overlap over a range of different meshes, recording tetrahedron overlap percentage (percentage of volume covered by a vesicle) when 0, 1, 2 or 3 vertices were covered (overlap for 0 is possible because vesicles can cross a face without covering a vertex, and overlap of 4 is always 100% so this doesn't need to be recorded). However we couldn't find any usable approximation. The plot below shows an example with vesicles approximately 40 times larger than tetrahedrons, and different size meshes (and therefore different volume ratios) showed similar pictures. So, while we agree this was worth investigating, we decided we couldn't reasonably apply any approximation - all overlaps showed a wide range and cannot be reduced to a simple proportion. We should also note that the overlap calculation is not usually a significant bottleneck in the simulation so this would bring limited benefit anyway.

We have expanded on the typical size of vesicles, tetrahedra and time steps in the manuscript- in particular in Figures 2, 3a and supporting text.

Yes, it is a list of species at a given number but, in the context of raft formation, present in a triangle. We have added this definition (**lines 414-416**).

Specific comments/suggestions

Referee comment	Author reply									
8. Line 436: regarding “obviously”, please clarify this point a little:  • what happens if the step size is too big? • will the contents of tetrahedrons not evacuate quickly enough once the vesicle moves on top of it? The rest of the manuscript implies that the contents are suddenly moved to neighboring (not covered) voxels, but this may not be obvious to everyone. • might the simulation break or become inaccurate with bad voxels sizes, where total covering happens a lot? 	On reflection, this is not really an “obvious” point and so we have moved this poor choice of words and have provided greater discussion on this topic. While we intuitively believe that it is ideal for vesicles to diffuse lower than one diameter per step so that it continuously overlaps some volume as it moves (no gaps), it is quite difficult to define exactly what is a good step size and when is a step too big. Brownian diffusion will still be modeled and diffusion rates still modeled accurately with large steps sizes. However, we do suggest in a newly added section (Sources of error, and reporting to user: Vesicle diffusion) (from line 606) that vesicles should diffuse most of the time (we chose 95% of the time for reports to user) at a distance that is less than their diameter plus the smallest diameter of other vesicles, which prevents them from potentially missing obstacles. In addition, as described in Sphere-Tetrahedron overlap, a diffusion distance smaller than the vesicle’s diameter allows for a faster overlap search, so is a good choice for practical reasons (lines 518-521). To answer the specific point about covering with large step sizes: the simulation will not break and it doesn’t really matter if total covering happens a lot- volumetric species will still be assigned effectively instantaneously to nearby non-covered voxels and will be available for SSA processes. We were not able to find any significant effect on our validations of reaction rates with large diffusion steps, other than when cytosolic reactants are immobile (as shown in new Figure 8c and discussed with supporting text, lines 662-683 and Supplementary Note 13).									
9. Fig. 6 suggests that vesicles move their entire diameter in one step, that seems fast, and contradicts the previous statement that they should move 10 nm per step. If this is because the illustration works better this way, then maybe it needs some minor adjustments with gray in-between circles of vesicle positions.	Thank you for pointing this out. While we do prefer to keep the large diffusion distances shown in this figure (now Figure 9) for clear illustration purposes of vesicles changing partition, we have added a note in legend explaining that this is just illustration and that real diffusion distances will often be smaller.									
10. Line 64: include a citation/reference for docking, priming, and fusion.	We have added a reference here, as requested (now line 63).									
11. Line 173: “user defined speed” (here and at other places within this section and suppl. note 4 and 5), please clarify if the dwell time is included when you speak about speed or not.	Yes, it is- sampled speed is effectively the distance traveled over the summed dwelltimes, and captures the user-defined speed (with noise originating from the stochastic jumps- see next point). We have added clarification to the main text (now lines 211-213) and to Supplementary Note 3.									
12. Line 190: please clarify the statement in parentheses. What step-size and stochastic effects are you referring to?	Based on the previous comment, we have added statements to both the main text (lines 209-213) and Supplementary about the origin of stochastic effects here. It is because of sampling the dwelltimes from a distribution, and not a single number, that there is noise on the dwelltimes and therefore there are noticeable stochastic effects in the “walk” - as can be seen in Supplementary Movie 1, for example. We remove this effect by taking many small steps for our validation purposes, at a step size of just 0.1nm. This value has now been added to Supplementary Note 4.									
13. Line 370 : these four combinations of reversible and reaction order could form a table     first order second order     irreversible Fig. 5b Fig. 5e   reversible Fig. 5c Fig. 5f   		first order	second order	irreversible	Fig. 5b	Fig. 5e	reversible	Fig. 5c	Fig. 5f	Thank you for this suggestion, which we considered, although we concluded that we prefer to keep the descriptions in text and the caption.
	first order	second order								
irreversible	Fig. 5b	Fig. 5e								
reversible	Fig. 5c	Fig. 5f								
14. Line 387 : Maybe split this long sentence into two; suggestion: We tested our initial MPI implementation in a realistic biological scenario. In this stringent test, we achieved a good reduction [...]	We have reduced this lengthy sentence. Since the model is defined in the previous line, we simple start with “In this stringent test...” (line 427)									
15. Line 437 : Include the symbols of the different quantities (diffusion coefficient, time step, diffusion distance) already here like this: “[...] for a diffusion coefficient D of $\text{0.11}\{\square\text{micro}\text{metre}\per\text{second}\}$ [...]” or similar.	Thank you for this suggestion. Although this particular line doesn’t exist any more, we have changed all quantities and units throughout the manuscript accordingly.									
16. Line 454: “If a vesicle is diffusing [...]”, please reformulate. Maybe with: diffusion distance smaller than the vesicle’s diameter.	We have reformulated this (now lines 518-519).									
17. Line 489: “as shown in as shown in” is doubled.	We have fixed this, thank you.									

18. Line 438: “0.11 $\mu\text{m}^2/\text{s}$ ” and “10 nm” diffusion distance. Is this correct? Please double check. We get roughly 0.8 nm: <pre>D <- 0.11e-12 # m²/s dt <- 1e-6 # s dr <- sqrt(6*D*dt) # 8.124e-10 m</pre>	We have removed this line and now have an expanded section on diffusion distances, as shown in Figure 3a for time-steps of 0.1ms and 1ms for a vesicle diffusing at $0.06 \mu\text{m}^2 \text{s}^{-1}$.
19. Line 442: perhaps, in one location early on use the term Brownian motion in connection with these equations. (can be a footnote)	We have added the term Brownian motion in connection to these equations (now line 495).
20. Line 459: “[...] a walking algorithm [...]”: this is a little unclear. We can imagine what it means, but perhaps we are mistaken: you start in a random or user specified location and make Gaussian updates to the current location until a valid location is found.	We added a description of the walking algorithm (now lines 522-535).

Comments regarding Supplement

Referee comment	Author reply
21. Note 4 & 5: Please mention step sizes , dwelt-times (or possibly omission of), and state whether the mean speed takes dwell-time into account.	The requested information has been added to Note 4. Note 5 has now been deleted due to space constraints (the panel previously described by Note 5 has been removed from Figure 3) and because this was not really validating the implementation.
22. Note 6: Was the simulation environment a sphere with 2d-diffusion, where diffusion happens on the edges between triangles? Or was it a thin volume (like a peel). I.e. is it literally 2D or effectively 2D. Please state explicitly.	It is 2D diffusion between triangles. We have added this description to the Note (now Note 5).
23. Note 6: Regarding iterations : please clarify whether these are time-step iterations, or repeated simulations with different random number seeds (a statistical ensemble)?	This is repeated simulations with different random number sequences, although we don’t actually reseed- we continue with the random number generator from the end point of the last simulation. We have made this clear now in the Note (now Note 5).
24. Note 6: What is naïve planar diffusion with angular displacement on a vesicle? Please either expand or omit entirely.	We have omitted this (now Note 5).
25. Note 9: Fig. 2b should be Figure 4b. In paragraph starting with “This model [...]”, below equation (6), “reversible” contradicts the previous statement above equation (6) where it is labeled “irreversible”.	Thank you, we have corrected these two mistakes (now Note 8). The figure is now Figure 5b, because we have added a new Figure 2.
26. Note 10: We suggest to change “because only one type of species was involved” to because both reactants are of the same species, their initial concentrations are identical and we can use eq (6) (get to the point about equal initial concentrations).	We have changed this as suggested (now Note 9).
27. Note 12: suggest to remove the word “iterations” at the end to not confuse with discrete time-steps (which are iterations within one simulation). Generally, once you have iterations within iterations or models within models, more precise language can help a lot. But, perhaps these iterations are time-steps and we misunderstood.	As above (point 23) we have replaced the word “iterations” with a clear description of what we do, here and at all other places in the document. It is repeated simulations with different random number sequences.

Notes on spacing, typesetting, and LaTeX

These are suggestions, or very minor corrections, listed by line:

Referee comment	Author reply
28. line 30 Is the section title “Main” deliberate?	This should have been changed from the generic Nature formatting before submission to Communications Biology. We have corrected this mistake and changed the section title to “Introduction”. Thank you for pointing this out.

29. line 154 Change the math μ to \upmu for an upright μ. Or, even better: <pre>\usepackage{siunitx} ... \unit{\square\micro\metre\per\second} \qty{36}{nm}</pre> using the <code>siunitx</code> package. Upright μ <code>\upmu</code> is from the <code>upgreek</code> package (on page 438 there is an upright μ, so the authors are probably well aware). The command <code>qty</code> from <code>siunitx</code> also puts a half-space between number and unit and allows the user to switch between different error notations (concise, or \pm). Same μ inconsistency in the caption for Fig. 2. In the same caption μ is instead a latin <code>u</code> in <code>0.5um²/s</code>, which is sometimes permitted as an ASCII replacement for μ (<code>\qty{0.5}{\micro\metre²/s}</code> would work as well if you want to type less).	Thank you for this suggestion. It is indeed better to use a standard package for units, and so we have modified all units in the main text to use <code>siunitx</code> accordingly.
30. line 174 <code>\sim</code> (tilde) should perhaps be <code>\approx</code> when approximate values are given. Perhaps even better: use ranges and error notation, so that it is clear to everyone how approximate those values are. On line 351 a range is given as “~50nm to hundreds of nanometers”, here it could be better to write tens/dozens/several to hundreds of nanometers or between approx. 50 nm and 1000 nm (or whichever numbers seem reasonable). The cited publication, [63], does a very similar thing, so this is probably fine, but they don’t write it in the same sentence.	We have changed the text here and use now <code>\qtyrange</code> to give a specific range. We quote a 2006 conference report where a range of 10-200nm was adopted, and a more recent FRET-based study that measures a smaller lower bound of 5nm (now lines 388 to 390). We also use <code>\qtyrange</code> now for the sizes of dense-core vesicles (lines 485-486). In addition, we have removed the other approximate value that existed for the myosin step-size of 36nm (line 208). After a thorough review of the literature we find that this number is usually not reported as an approximate value, perhaps because values outside of this number would cause the motor protein and cargo vesicle to spin around the helical structure of the actin filament as it walks, and we follow this convention. We cite the first paper that describes the 36nm step as a 25nm working stroke and 11nm diffusive movement (line 209).
31. line 174 <code>myosinV</code> should be Myosin V, probably.	We have fixed this, thank you (now line 208).

REVIEWERS' COMMENTS:

Reviewer #1 (Remarks to the Author):

N/A

Reviewer #4 (co-reviewed the manuscript with Reviewer #3) (Remarks to the Author):

We agree that all of our points and comments were fully addressed. A lot of care was put into the response.

We see no necessity to correct or improve anything else. But the authors may fix little typesetting peculiarities (see below), without the need to re-examine this through review.

The new figures are very good and clear up the properties of the modeled objects in relation to one another.

In the new line of the caption for figure 3 (a), the word STEPS refers to the discrete updates to a vesicle's position rather than the software, unless we are mistaken. In that case, it shouldn't be in capital letters.

The new Figure 7 is very information dense, but works well with the Caption and is very helpful. Please note that "Point__species" seems to have two spaces between the words (or looks like it), thus making the word "species" visually closer to the green box to the right of it.

Is our interpretation correct that a raft can contain an anti-dependency of Raft surface reactions (optionally) and in that case nothing would be consumed or produced? So, this column combines both cases in one picture: success and possible failure of reaction. It is perhaps the only column to do so and thus sticks out a bit conceptually.

The new Section on error detection and reporting is very informative and reassuring. Thank you for going into this in detail.

We thank the authors for investigating the vesicle overlap and the number of covered vertices as a candidate for a ball-park approximation. It's good to know that these calculations are not a bottle-neck.

The expanded Supplement is helpful and we appreciate Table 1.